# Dynamic Rainfall Erosivity Estimates Derived from GPM IMERG data

Robert A. Emberson[1,2]

[1]Hydrological Sciences Laboratory, NASA Goddard Space Flight Center, Greenbelt, MD, 20771, United States
[2]GESTAR-II, University of Maryland Baltimore County, Baltimore, MD, 21250, United States

*Correspondence to*: Robert A. Emberson (robert.a.emberson@nasa.gov)

**Abstract.**

Soil degradation is a critical threat to agriculture and food security around the world. Understanding the processes that drive
soil erosion is necessary to support sustainable management practices and to reduce eutrophication of water systems from
fertilizer runoff. The erosivity of precipitation is a primary control on the rate of soil erosion, but to calculate erosivity high
frequency precipitation data is required. Prior global scale analysis has almost exclusively used ground-based rainfall gauges
to calculate erosivity, but the advent of high frequency satellite rainfall data provides an opportunity to estimate erosivity using
globally consistent gridded satellite rainfall. In this study, I have tested the use of GPM IMERG rainfall data to calculate global
rainfall erosivity. I have tested three different approaches to assess whether simplification of IMERG data allows for robust
calculation of erosivity, finding that the highest frequency 30-minute data is needed to best replicate gauge-based estimates. I
also find that in areas where ground-based gauges are sparse, there is more disparity between the IMERG derived estimates
and the ground-based results, suggesting that IMERG may allow for improved erosivity estimates in data-poor areas. The
global extent and accessibility of IMERG data allows for regular calculation of erosivity on a month-to-month timeframe,
permitting improved dynamic characterisation of rainfall erosivity across the world in near-real time. These results demonstrate
the value of satellite data to assess the impact of rainfall on soil erosion and may benefit practitioners of sustainable land
management planning.

## 1 Introduction

Topsoil is a key component of the Earth's critical zone, acting to sequester carbon, filter pollutants from water, and supporting
growth of plants. Agricultural topsoil in particular is the fundamental basis upon which food security relies, and sustainable
management of topsoil is one of the defined UN sustainable development goals [UN, 2015]. However, land management
practices around the world have led to significant degradation of topsoils, with global analyses suggesting that the "majority
of soils are in only fair, poor, or very poor condition" [FAO, 2015]. Soil degradation threatens communities around the world
with food insecurity in the next decades, alongside limits to water supply for irrigation [Hanjra & Qureshi, 2010]. Across large
parts of global agricultural zones, the rate of topsoil loss far exceeds the replenishment rate [FAO, 2015]. Soil loss costs

hundreds of billions of dollars each year [GSP, 2017] and given that it may lead to significant declines in productive agricultural land area by 2050, preservation of topsoil is essential to ensure a sustainable future [Steffen et al. 2015].

Degradation of soil is driven by many factors including cover by impermeable materials, physical compaction, wind and rain-driven erosion, salinization and chemical degradation (Ferreria et al. 2018). Of these, liquid precipitation is a major driver of soil erosion. Rainfall washes off loose soil and determining the erosivity of rainfall is a major component of soil loss calculations. The widely used Revised Universal Soil Loss equation [Renard, 1997, USDA, 2013] incorporates rainfall erosivity as the main dynamic factor determining soil loss. Calculating and measuring rainfall erosivity is therefore imperative to support models and observations of soil degradation. Some global scale analyses have relied upon ground-based rainfall gauges to provide observations of rainfall intensity and duration [Panagos 2015, Borelli et al. 2016, Yin et al. 2017], including the widely used Global Rainfall Erosivity Database (GloREDa, Panagos et al. 2017), but there is major geographic variability in the availability of rainfall gauge data around the world [Panagos et al. 2017], and gauge sparsity is particularly pronounced for gauges that record at sufficient temporal frequency to characterise erosivity.

Gridded rainfall data represents an alternative to interpolating gauge data, with several recent studies employing national or global scale gridded precipitation data. Padulano and coauthors [2021] use ERA-5 reanalysis rainfall data and European Gridded rainfall data (E-OBS) to calculate erosivity over Italy, Matthews et al. [2022] tested other datasets including EMO-5 and UERRA MESCAN-SURFEX data across Europe, and Raj and coauthors [2021] used the Indian IMDAA data to estimate erosivity across India. Reanalysis data was also employed globally by Bezak and coauthors [2020], and on the Tibetan plateau by Chen et al. [2022]. These studies have demonstrated the value of a more consistent gridded dataset, which can help limit errors from interpolation of widely spaced gauges. However, many of the global-scale reanalysis datasets are only available at hourly (e.g. ERA-5) or lower frequency, creating potential issues for erosivity estimation which is benefits from high-frequency rainfall data.

Recent studies have utilized gridded satellite rainfall data to provide global spatially consistent estimates of erosivity, offering an alternative to other gridded data to help with filling in gaps in areas that are sparse in terms of ground-based gauges. Bezak and coauthors [2022] have used data from the National Oceanic and Atmospheric Administration (NOAA) Climate Data Record (CDR) Climate Prediction Center MORPHing (CMORPH) dataset, which offers global scale data at a 30-minute resolution. Li et al. [2017] utilized the Tropical Rainfall Measurement Mission (TRMM) data to estimate erosion, but this data is only available at 3-hour resolution, which is not widely considered to be sufficiently high frequency to reliably estimate erosivity.

In this study, I use the 30-minute data from the NASA IMERG (Integrated Multi-satellitE Retrievals for GPM (Global Precipitation Mission)) dataset, which covers the time period from 2000 until present, to estimate rainfall erosivity at a global scale. I have also tested several approaches to calculate erosivity from rainfall-time series, and the comparison with the ground based GloREDa dataset [Panagos et al. 2017] provide insight into the value of high temporal frequency rainfall data for erosivity estimation. This application of IMERG data provides an additional method for calculating erosivity at a global scale and allows for dynamic estimates of erosivity at various time intervals, depending on the calculation method chosen.

## 2 Methods

### 2.1 Rainfall Data

In this study, three methods employed in previous studies are employed to calculate rainfall erosivity using the IMERG rainfall dataset. IMERG version 6B is the latest version of the long-running IMERG rainfall dataset [Huffman et al. 2019]. IMERG
utilizes observations from several different satellites to estimate precipitation across most of the surface of Earth. The IMERG algorithm uses microwave observations from the Global Precipitation Monitoring (GPM) satellite (2014-present) and also the earlier Tropical Rainfall Measuring Mission (TRMM) satellite (2000-2015). Microwave observations are included from a range of other satellites, and a series of merging methods are used to interpolate between these observations [Joyce et al. 2011, Huffman et al. 2011, Hong et al. 2004], including the CMORPH-KF algorithm [Joyce et al. 2011]. Infra-red observations are
also used to support the interpolation of results [Huffman et al. 2019]. While a near-real time product is available, the research grade IMERG 'final' uses data from the Global Precipitation Climatology Center (GPCC) ground-based gauges to correct the initial 'IMERG-Late' results. In this study, I use the 'final' IMERG v06B product, at three different temporal resolutions: monthly, 3-hourly, and the highest frequency 30-minute product. All three of these are available from 2000-present, but since the algorithm processing starts after January 1$^{st}$, 2000, I have started the analysis from January 1$^{st}$, 2001-December 31$^{st}$, 2021.
IMERG has a grid cell of 0.1 decimal degrees (approximately 9km at the equator) and so all erosivity outputs maintain this grid cell size.

IMERG v06B does distinguish between liquid and non-liquid precipitation (rain and snow), and I use the estimate of liquid precipitation proportion to calculate only the liquid precipitation. IMERG-Final data includes a specific field that estimates the fraction of liquid precipitation in each grid cell for each data point. The liquid precipitation was therefore calculated by
multiplying the total precipitation field by the fraction of liquid precipitation field. Although across most the latitude range studied here (60 degrees N – 60 degrees S) rainfall, rather than snow, is the dominant precipitation type, but since rainfall is the key type of precipitation that determines erosion of surface material I suggest it is important to perform this correction to exclude non-liquid precipitation when calculating rainfall totals.

### 2.2 Calculating Rainfall Erosivity

Rainfall erosivity depends on both the intensity and duration of rainfall in a given location. The United States Department of Agriculture (USDA) has developed the Revised Universal Soil Loss Equation (RUSLE) which utilizes rainfall erosivity as a key factor [USDA, 2013]. The relationships developed for RUSLE have been widely tested and are considered the standard set of equations to determine erosivity from ground-based data. The most recently revised version of the RUSLE model
[USDA, 2013] converts rainfall into erosivity using the following series of steps. First, the rainfall time-series is divided into specific storm events, each separated by periods of 6 hours or greater where rainfall was less than 1.27mm. Rainfall events with less total rainfall than 12.7mm (0.5 in) are excluded from calculations [Brown and Foster 1987]; this was initially to

reduce computational load, but studies have shown that rainfall events with lower totals than this do not significantly contribute to overall erosivity.

Once the rainfall events have been isolated, the specific kinetic energy $e_k$ (units of MJ ha$^{-1}$ mm$^{-1}$) is calculated. Different studies have used a variety of coefficients for this equation; in this study, I have used the current USDA RUSLE 2 model coefficients [USDA, 2013]. I have also tested the older RUSLE 1 coefficients for comparison with older studies for one of the rainfall erosivity calculation approaches, with limited differences observed. The RUSLE 2 equation is as follows:

$$e_k = 0.29.[1 - 0.72 . e^{(-0.082 . I)}] \qquad \text{(Equation 1)}$$

Where $I$ is rainfall intensity in mm.h$^{-1}$. To calculate erosivity, first the total kinetic energy of the rainfall event is calculated:

$$E = e_k . I . \Delta t \qquad \text{(Equation 2)}$$

Where $E$ is the total kinetic energy and $\Delta t$ is the time interval in hours. The erosivity is then calculated:

$$R = \frac{\sum_n E . I_{30}}{N} \qquad \text{(Equation 3)}$$

Where $I_{30}$ is the maximum 30-minute rainfall intensity of rainfall event $n$, which occurred over a time span of $N$ years. R

therefore has units of MJ.mm ha$^{-1}$ h$^{-1}$ yr$^{-1}$.

Using these equations, and the separation of the rainfall record into storms as described above, I have calculated R from the 30-minute IMERG rainfall record from January 2001 until December 2021. This provides a single value for R in the 3600x1800 cells of the IMERG record, which I have then reduced to only global land areas between 60 degrees N and 60 degrees S, to exclude areas where the IMERG record is incomplete. Additionally, I have calculated the monthly values for R for the year

2020, to demonstrate the applicability of this method to estimate erosivity dynamically.

This approach has been successfully and widely used with ground-based gauge analysis of erosivity using the RUSLE model since 1987, but alternative approaches have also been developed. Earlier iterations of satellite rainfall products from the TRMM satellite constellation were available only at a 3-hour time interval, which precluded calculation of 30-minute intensity. Prior studies utilizing TRMM products to calculate erosivity have used two other methods; the Modified Fournier Index (MFI), and

by considering each 3-hourly rainfall window as an individual storm [Vrieling et al. 2010]. Because analysing 20 years of 30-minute IMERG data is computationally intensive (over 9TB of data are analysed in total), I have also tested these two methods using global IMERG data to assess compare their performance would warrant the use of these simplified approaches. I have also used the 30-minute IMERG data to calculate R according to equations 1-3 above. The details for the MFI and 3-hour storm approaches are described next.

The Fournier Index [Fournier, 1960] was an early model to describe rainfall erosivity that relied on only low-frequency recording of rainfall. It is defined as:

$$FI = \frac{p^2}{P} \qquad \text{(Equation 4)}$$

Where $p$ is the average rainfall of the month with the highest rainfall, and $P$ is the average annual rainfall. Arnoldus [1977] revised this index to create the Modified Fournier Index (MFI), defined as follows:

$$MFI = \frac{1}{P} \sum_{i=1}^{12} p_i^2 \qquad \text{(Equation 5)}$$

Where $p_i$ is the average rainfall in month $i$. Arnoldus [1977] demonstrated significantly better agreement between short-term rainfall observations of erosivity and the MFI values than FI values, and it has been used by other authors since [Renard and Freimund, 1994], including with satellite rainfall data [Vrieling et al. 2010]. The units for both of these indices are mm (mm$^2$/mm), and prior studies have demonstrated it is most effective when applied to homogenous climatic regions [Arnoldus 1977, Renard and Freimund, 1994]. To convert between MFI and R values, various strategies have been applied, with different coefficients derived for various climatic zones. Disagreement remains in the literature surrounding the appropriate equations to use, and which exact units were used by Arnoldus [1977] (see Majhi et al. [2022], Chen & Bezak [2022], Mahji et al. [2021], Renard and Freimund [1994]). In this study I have not converted the raw MFI values, since the intention is to contrast the relative correlations with the R factor estimates from higher frequency rainfall data and ground-based analyses. Given the relative limitations to the MFI calculation (see below), the alternative methods presented here may offer more promising solutions for future studies. Nevertheless, MFI values are widely used and still maintain some advantages, in particular given their relatively low computational level and applicability to more temporally sparse data. I have calculated MFI values both for the entire IMERG record, as well as the interannual variability.

Vrieling and coauthors [2010] tested an alternative method to calculate R using 3-hour satellite rainfall data, prior to the advent of the 30-minute IMERG rainfall data. The specific kinetic energy of rainfall is defined as in equation 1 (although Vrieling et al. [2010] use different coefficients). Because of the lower temporal resolution of the 3-hour data, under this approach it is not considered justified to define the start and end of a storm event in the rainfall records. As such, each 3-hour window is considered as a storm event, and the kinetic energy for each 3-hour window is calculated as such:

$E3h = e_k \times 3I$ (Equation 6)

With $I$ meaning the rainfall intensity, in mm.h$^{-1}$. To calculate R according to equation 3, the maximum 30-minute intensity of the storm is required, but since that information is unavailable with the 3-hourly data, Vrieling et al. [2010] used the average intensity of the 3-hour rainfall period as a multiplier. However, with the 30-minute IMERG data, I am able to calculate the maximum in each 3-hour window, so in this estimate, R is defined as:

$R = \sum_{j=1}^{N} E3h_j \times I30_j$ (Equation 7)

In equation 7, j represents the j$^{th}$ storm, from 1-N. As with the 30-minute version, the units are: MJ.mm ha$^{-1}$ h$^{-1}$ yr$^{-1}$, and as such these estimates are directly comparable with those of the 30-minute estimates, as well as ground-based estimates. Since several published estimates for the coefficients for specific kinetic energy are available, I have tested the RUSLE 2, RUSLE 1, and Vrieling et al [2010] coefficients for this estimate of R. This is because the lower computational requirement of this estimate allows for testing of multiple coefficients. These are discussed below.

**2.3 Ground-based comparison**

Rainfall erosivity analysis has typically been conducted at a local or regional scale, but fortunately recent work by Panagos and coauthors [2017] derived the first global scale erosivity estimate, the Global Erosivity Database (GloREDa). This model uses 3625 ground-based stations, with rainfall records averaging 17 years, to calculate a global estimate of erosivity based on

the equations described above (equations 1, 2 and 3) and using a Gaussian regression model to interpolate between stations. The density of stations significantly varies, with 48% in Europe, and only 5% in South America and Africa. I have used the GloREDa data as an independent comparison for the IMERG-derived erosivity estimates for this study. The GloREDa estimates are available at a higher spatial resolution (30 arc-seconds) than the IMERG derived estimates, so I have downsampled the GloREDa data to the native IMERG resolution (0.1 degrees) using a bilinear interpolation method. This avoids creating additional data from the IMERG-based approaches. I compare the results of the IMERG and GloREDa estimates both at a cell-by-cell scale but also in terms of overall statistics, including mean, median and standard deviation values.

## 3 Results

### 3.1 30 Minute Erosivity

The 20-year average erosivity values estimated using IMERG 30-minute rainfall data are shown in Figure 1, part A. The global map highlights critical hotspots for rainfall erosivity, as well as areas where low rainfall levels lead to very low erosivity. Significant areas where erosivity is elevated include the areas of Central America and the Northern part of South America, the Himalayas (with particularly high rates in the Indus-Yarlung Suture Zone at the Eastern Himalayan Syntaxis), the Indonesian Archipelago and Papua New Guinea, and Bangladesh. Low erosivity values are estimated for much of the global desert regions, with a broad belt of low erosivity spreading from the Western Sahara, across the Arabian Peninsula, through into Southern Siberia. An interesting emergent trend is for very high erosivity values in some coastal areas, including the Sub-Andean coast of Colombia; the Pacific North-West of the United States and Canada; the coasts of Guinea-Bissau, Guinea and Sierra Leone; the Western Ghats of India; and much of the Bangladeshi-Myanmar coast. Many of these coastal zones are impacted by infrequent but extreme tropical storm rainfall events, and it is possible that these contribute to the high erosivity values calculated in these areas.

### 3.2 Alternative Erosivity Estimates from IMERG

The IMERG-derived erosivity estimates calculated for the 3-hour and MFI storm approaches are shown in parts B and C of Figure 1. While the overall global patterns are broadly similar to those produced by the 30-minute version (Figure 1A), including the trend of high erosivity values observed in many coastal areas, the absolute values differ quite significantly. While the MFI values (Figure 1C) are not directly comparable to the R values calculated via the other metrics, prior studies have suggested a non-linear scaling relationship with exponents of ~1.5 [Arnoldus, 1977], the large spread of values observed when comparing cell-by-cell values for the MFI and 30-minute R values (Figure 2A) do not support a single, consistent scaling relation. Instead, this likely supports a scaling relationship dependent on local climatic conditions, as has been emphasized by other authors [Smithen and Schulze 1982, Renard and Freimund 1994].

The values obtained with the 3-hour storm model (Figure 1B) are directly comparable to the 30-minute version since they have the same units. As with the MFI values, the overall patterns that emerge globally are similar, but once again there are differences in absolute values. The Pearson correlation coefficient of the cell-by-cell values of the 30-minute and 3-hour estimates is high ($R^2 = 0.923$), but this is not a perfect correlation, nor an exactly linear relationship (Figure 2B). Particularly at high erosivity values, the estimates diverge to a greater degree, with the 30-minute model generally producing higher values. The linear estimate of the best-fit line between the two models is has a scaling relationship (slope) of 1.8, implying the 30-min versions are markedly higher. Although the 3-hour rainfall approach includes all 3-hour windows, rather than excluding smaller rainfall events (as described in the methodology above), the 30-minute approach captures rainfall events with larger short-term rainfall intensity, which will result in larger erosivity estimates since the scaling between rainfall intensity and erosivity is non-linear.

### 3.3 Comparison with Ground-Based Data

I have compared the results from the three different IMERG-based estimates of erosivity with the ground-based observation estimates from GloREDa [Panagos et al. 2017]. Since the IMERG analyses are not applied above 60 degrees N, the intercomparison is only for the IMERG region, although GloREDa covers the entirety of the Northern Hemisphere. Maps of the ratio of erosivity estimates derived from the 30-minute IMERG data, the 3-hour IMERG data, and the MFI estimate are shown in Figure 3 (section A, B, and C respectively). Note that the values shown for the 30-minute and 3-hour data are directly comparable to the GloREDa data (identical units), but the MFI estimate represents a different unit, and so Figure 3C is only appropriate to assess spatial patterns in the ratios, rather than absolute values.

Across all three IMERG methods, similar patterns emerge. There is a strong degree of agreement between GloREDa and the 30-minute and 3-hour IMERG estimates across much of Europe and Northern Asia, but much more marked differences elsewhere. In particular, GloREDa shows higher values across the Sahara, Central Asian Deserts, and the North American West. IMERG-estimates do however show clearly higher estimates in the immediate vicinity of extremely dry areas where no rainfall is recorded, in parts of the Northern Sahara and Arabian Desert. In wetter areas, differences between IMERG and GloREDa also emerge; significantly greater values in erosivity from IMERG are observed in the several coastal areas, including the Western Ghats of India, the coast of Bangladesh, Myanmar, and Thailand, and the Pacific coasts of Colombia and British Columbia in Canada.

To simplify the comparison between each of the datasets, I have plotted the probability density functions of the cell values for each in Figure 4. Of the IMERG-based assessments, the 30-minute estimate most closely matches that of GloREDa. Even if the MFI values are normalized by the maximum values (Supplementary Figure 1), they still do not provide a close approximation of the GloREDa values, with a much lower variability in the values. In Figure 4, both the 30-minute and 3-hour estimates both show a similar peak around the modal values as the GloREDa, but the second peak in values is somewhat lower in both IMERG-based estimates than GloREDa. The 30-minute output also shows a longer tailed distribution, with a greater

proportion of values at very low erosivity and some at higher erosivity than GloREDa. In essence, the 30-minute model produces more low-erosivity estimates than GloREDa, and while the most commonly-observed erosivity values (between 200-600 MJ.mm ha$^{-1}$ h$^{-1}$ yr$^{-1}$) are broadly similar to GloREDa, there is a larger degree of disagreement at higher erosivity values. I

discuss the possible reasons for discrepancy in section 4, below. It is notable also that there is a large degree of difference in the cell-by-cell values even within different continents (Supplementary Figure 2), when they are compared against one another; there is significant dispersion between all of the IMERG-derived estimates and the GloREDa values. The Pearson R-squared values for each are shown in Table 1, while continent-by-continent values for mean, median and standard deviation for the 30-minute IMERG data and GloREDa values are shown in Table 2.

| Comparison pair | Pearson R-squared value | Slope |
|---|---|---|
| GloREDa – 30-minute IMERG | 0.498 | 0.56 |
| GloREDa – 3hr IMERG | 0.633 | 0.34 |
| GloREDa - MFI | 0.656 | 0.02 |

Table 1: Pearson R-squared values for the comparison between GloREDa and IMERG-based estimates for erosivity.

| Continent | 30-min IMERG mean | 30-min IMERG median | 30-min IMERG Standard deviation | GloREDa mean | GloREDa median | GloREDa Standard deviation |
|---|---|---|---|---|---|---|
| N. America | 839.2 | 361.9 | 1358.9 | 1676.3 | 744.8 | 2072.3 |
| S. America | 3367.1 | 2656.4 | 3138.4 | 5895.4 | 6266.3 | 3361.5 |
| Europe | 484.1 | 350.8 | 525.6 | 550.5 | 402.9 | 414.4 |
| Asia | 1313.6 | 153.5 | 2978.1 | 1856.7 | 398.6 | 2927.3 |
| Africa | 1231.5 | 708.3 | 1487.2 | 3356.8 | 2619.3 | 2977.6 |
| Australia | 659.5 | 215.7 | 1099.4 | 1533.9 | 950.7 | 1596.2 |
| Oceania | 4802.1 | 2375.7 | 5127.0 | 4100.8 | 2254.1 | 4539.9 |

Table 2: Statistics of the erosivity estimates for each continent for the GloREDa and 30-minute IMERG data. Units for all values are MJ.mm ha$^{-1}$ h$^{-1}$ yr$^{-1}$

Since GloREDa is an interpolated dataset based on gauge-derived estimates of erosivity, the specific values in a given grid cell will not always represent the exact gauge-derived value for a given pixel. To account for this, and to compare the IMERG-derived values with those of ground-based gauges used to calibrate GloREDa. The Rainfall Erosivity Database on the European Scale (REDES, Panagos et al. 2015) is an openly available dataset of gauge-derived estimates of erosivity. In Figure 5, the values from the IMERG-based estimates for Europe are shown in comparison with the values from the REDES dataset.

Spatially, the IMERG based analysis performs well in several European countries, including Greece, the Iberian Peninsula,

East Germany, France, Switzerland and parts of Italy. IMERG overestimates in comparison with gauges in the Western parts of the United Kingdom and Ireland, and to some extent in Western Germany and the low Countries, while underestimating in Hungary and parts of Bulgaria and Romania. Given that the gauges that make up REDES are not uniformly distributed, a statistical comparison of the two datasets will be dominated by countries with higher gauge density (like Belgium, Italy, and Slovakia). The slope of the relationship between the two datasets is 0.26, while the slope of the relationship with GloREDa is 0.5 – although given that this dataset is used to calibrate GloREDa, this is not unexpected. Although the spatial patterns in Italy are reproduced by the IMERG data, the values obtained are in some cases an underestimation across much of Italy, which has a very high number of gauges represented in the REDES dataset. IMERG and other satellite rainfall datasets have lower accuracy in topographically complex settings, and worse performance of IMERG in capturing intense rainfall in the mountainous parts of Italy and the Carpathians may be one of the reasons for the underestimation of the IMERG-based erosivity estimates, although further research and comparison with ground-based gauges is certainly warranted. Recent research has shown that satellite rainfall datasets, including IMERG, may consistently underestimate the total amounts of heavy and storm rainfall [Marc et al. 2022, Chen et al. 2023]

### 3.4 Monthly Estimates

The month-by-month estimates for erosivity for the year 2020 using the 30-minute data are shown in Figure 6, separated by continent. The violin plots in Figure 6A show the range of the data, as well as the mean (red bars) and median (black bars). Given the large size of each continent, there is a significant degree of variability across all continents; in Europe and Australia show lower mean and median values across the year than the other continents, but across the entire dataset there is enormous variability in each month, and only limited variability is observed from month-to-month as a result of seasonal rainfall variability.

Since the scale of each continent is so large that interannual variability may be difficult to observe, I have subset the monthly data to areas where cropland is present. Using the cropland data of Ramankutty et al. [2008], I have selected only the cells from each continent where the cropland proportion exceeds 80% (Figure 7). Much more significant variability is observed on a month-to-month basis, with larger peaks in erosivity observed in Africa and Asia in June-July-August, in South America from January to March, and smaller peaks in erosivity in North America and Europe. Note that there is insufficient data in Oceania with cropland higher than 80% to show statistics. By subsetting to areas with significant cropland, I highlight the impact of erosivity on agricultural areas and the months of the year where erosivity is of greater concern for farmers and planners across those continents.

### 4 Discussion
### 4.1 Divergence from Ground-Based Data

It is clear that although there are some areas, particularly in Northern Europe, where the 30-minute IMERG-based estimates and those of GloREDa broadly agree, there are large areas of the world where the ratio of the two estimates remains well under 1:1. Moreover, there are a range of other areas – in particular several coastal areas, where the IMERG-based estimates are much larger than those of GloREDa. It is likely that multiple systematic effects lead to these differences, and it is informative to examine the divergence between the two estimates in more detail to assess the robustness of the IMERG-derived estimates

of erosivity.

While the IMERG-based estimates use the coefficients for erosivity defined by the USDA [2013] for the updated version of RUSLE, GloREDa uses the earlier coefficients. I have calculated the 3-hour erosivity estimate for both equations to test whether this would lead to major systematic differences. The difference in coefficients does not lead to large divergences, and the two estimates are extremely highly correlated (Supplementary Figure 3). The updated RUSLE equation gives values that

are slightly larger than the earlier equation (approximately 1.1x larger). With all else being equal, this would suggest the 30-minute IMERG data should give a value of approximately 1.1 times larger than GloREDa; however, the IMERG results are in fact systematically lower (global slope: 0.56). As such, although this might explain some of the dissimilarity between GloREDa and the 30-minute IMERG values, it does not explain the overall lower values of the IMERG estimates. Since research has demonstrated that the coefficients in RUSLE version 2 better match independent observations of erosivity in contrast to the

RUSLE version 1 equations [McGregor et al. 1995], I consider it a justified approach to calculate erosivity in this study.

There are large regional differences in the ratio of the 30-minute IMERG estimates and the GloREDa values. In Figure 8, I have plotted the spread of values for each continent for the two estimates, with the red line indicating the line-of-best fit, and the black line indicating the 1:1 line. It is notable that continent where the values most closely fit the 1:1 line is in Europe,

followed by Asia. Elsewhere, the IMERG estimates are significantly lower. The GloREDa estimates are derived from a global set of rainfall gauges, but the highest density of gauges by a significant degree is in Europe [Panagos et al. 2017]. The higher degree of agreement between the IMERG and GloREDa estimates for erosivity are found in the areas with the highest density of gauges – in other words, where the ground-based estimates have the highest degree of calibration and validation. IMERG does use the Global Precipitation Climatology Center Gauges to calibrate the satellite-derived estimates [Huffman et al. 2019],

which has a higher density of gauges in North and South America, Africa, and Southeast Asia, than the gauges used in the GloREDa analysis [Panagos et al. 2017]. While both IMERG and GloREDa use spatial interpolation techniques, IMERG weighs the satellite inputs more heavily in areas where gauge density is lower (like Africa and South America) whereas GloREDa does not. Given that the two estimates have better agreement in terms of absolute values where GloREDa has the highest gauge density, I suggest that the disagreement elsewhere may be due to the lower amount of available data from

ground-based sources, whereas the satellite data provides a globally consistent estimate, which may be more robust for calculating erosivity.

In several desert areas around the world, including the Atacama and Namib deserts, GloREDa values exceed 100 MJ.mm ha$^{-1}$ h$^{-1}$ yr$^{-1}$, but given in some of these areas annual rainfall is lower than 10mm, the high values in GloREDa are physically

implausible. While some studies have shown IMERG v06 can over-predict rainfall in the Arabian desert (Alsumaiti et al. 2020), the exclusion of storm events with less than 12.5mm of rainfall in total would also exclude the small systematic error from satellite overprediction of very low rainfall totals in desert zones from influencing erosivity estimates. This further supports the use of IMERG over sparse-gauge based estimates for erosivity. Conversely, IMERG may miss highly local peaks in orographic precipitation – one example is the island of Maui in the Hawai'ian island chain, where local maxima in rainfall can exceed 5000mm of annual rainfall; however, this is not captured by the coarse IMERG data. As such, the satellite-derived estimates may be out-performed by gauge-based estimates where gauge density is very high or able to capture localised maxima.

It is important to note that GloREDa is also an interpolated dataset, and as such may have inaccuracies if gauge density is low. When directly compared with the REDES gauge-based dataset, the IMERG data reproduces some of the spatial patterns but clearly has other limitations. The significant overestimation of erosivity in the coastal Atlantic areas of the UK, Ireland and Portugal (Figure 5) supports the analysis of prior work (e.g. Tian and Peters-Lidard 2010) that shows that satellite rainfall products have larger uncertainties in coastal regions, and as such the use of IMERG-derived results in such areas may not provide accurate estimates in these areas.

As well as continent-by-continent differences, there are other clear zones of divergence between the IMERG and GloREDa estimates. As mentioned above, these include several coastal areas in India, Southeast Asia, and the Pacific coastlines of Colombia and Canada, where the IMERG estimates exceed those of GloREDa by a significant degree [Figure 3]. Although these areas may be subject to the coastal biases associated with all satellite rainfall products, these areas are all areas where both IMERG (Supplementary Figure 4) and GPCC-gauged rainfall is extremely high [Schneider et al. 2014], and the IMERG-derived erosivity estimates broadly mirror the spatial patterns observed in the annual gauged rainfall totals. Further research is needed to determine the erosivity of rainfall in these areas with lower gauge density to determine whether satellite-based estimates can be relied upon.

## 4.2 Limitations of IMERG-based erosivity estimates

Although IMERG provides a globally consistent estimate of rainfall, limitations remain both with the rainfall data and the calculation of erosivity. Since satellite observations of intense rainfall depend upon the satellite overpasses coinciding with the time of the local rainfall, it is possible that large, and particularly intense short peaks in rainfall (particularly when associated with infrequent tropical storms) may be missed by satellite rainfall products [Marc et al. 2022]. Although erosivity of rainfall does not scale in a strongly non-linear manner with rainfall intensity, these extreme storms may contribute a large proportion of overall annual rainfall in some settings [Khouakhi et al. 2017, Marc et al. 2022], so if these rainfall peaks are missed by the satellite observations, then rainfall erosivity may still be underestimated. It is notable that the IMERG-derived results are lower than the gauge-derived results in a number of locations and are lower on average in all continents except Oceania (Table 2), and so underestimation of rainfall events driving erosivity may be a reason for this. Bezak et al. (2022) highlighted that the largest 11% of rainfall events contribute 50% of the erosivity, so it is particularly relevant not to miss these very large events.

Since IMERG may miss very short-lived rainfall events, it is especially important if rainfall driving erosion is from extremely short-lived rainfall events, rather than longer storms. To explore this, I have analysed storm histories from four locations

around the world; two in areas of concern for soil erosion (Near Wichita, USA, and Lucknow, North India) one in a critical region of degradation where the IMERG estimate exceeds GloREDa (Central Sierra Leone) and near San Pedro de Atacama, in the dry desert of Northern Chile. In Chile, only 3 rainfall events are observed over the entire 2000-2021 interval. In the other locations, I tested what proportion of the storm events in each location is formed by the 30-minute period of rainfall and compared that to the total storm rainfall. Storms where the most intense short bursts of rainfall make up most of the total

rainfall are likely to be more underestimated by IMERG in comparison with storms where more consistent rainfall is observed. In Supplementary Figures 5A-D, I show the fraction of total rainfall in each storm from the 30-minute peak rainfall vs the cumulative kinetic energy from rainfall. In Sierra Leone and Lucknow, more than 80% of rainfall kinetic energy is derived from storms where the 30-minute interval of peak intensity is less than 50% of total rainfall. In Wichita, the storms are more dominated by the short term intense rainfall – 80% of kinetic energy is derived from storms where the maximum 30-minute

rainfall forms up to 80% of total storm rainfall. I suggest that in locations like Wichita, IMERG may be more subject to missing short-term bursts of rainfall. This may explain why IMERG is lower than GloREDa in Wichita and the US South East.

    In addition, the spatial resolution of IMERG data is large (0.1 decimal degrees), and so observations of local variability in rainfall totals as a result of orographic boundaries or other microclimatic differences will be limited by resolution. However,

agricultural zones with relatively low topographic variability are more likely to be represented fairly, and these zones may be more critical for the socio-economic impacts of soil degradation by rainfall erosion. I have compared the 30-minute IMERG-derived erosivity estimates with the ground-based estimates from GloREDa and plotted the ratio of the two estimates in comparison with the maximum topographic slope calculated in each grid cell (Supplementary Figure 6). The maximum slope is calculated from the NASA SRTM data [Farr et al. 2007]. There is not a significant change in the difference between IMERG

and GloREDa as slope increases, which suggests that slope does not significantly control the differences between satellite and ground-based estimates.

    As noted above, there are limitations with the simplifications used to generate the MFI and 3-hour erosivity estimates. The MFI value is only appropriately applied to climatically homogenous zones, which suggests a global MFI value is unlikely to be appropriate; moreover, the conversion from MFI to R-factor is neither consistent nor agreed upon in the published literature.

Despite the lower computational requirements to calculate MFI, I suggest that the limitations associated with it mean that the 30-minute version should be considered superior and used wherever data and computational capacity is available. While there is a good degree of agreement between the 3-hour estimate and GloREDa, prior research has shown that the 3-hour simplification of TRMM-era satellite rainfall data may reduce the accuracy of the results [Vrieling et al. 2010], and the 30-minute data is the only data source that can provide the appropriate temporal resolution [Bezak et al. 2022].

Other satellite-derived rainfall products are available, including GSMaP [Kubota et al. 2007, 2020] and the hybrid MSWEP [Beck et al. 2017], but since GSMaP is available only at a 1hr temporal resolution and MSWEP at 3hr maximum temporal

resolution, neither of these products are appropriate to calculate rainfall erosivity. Bezak and coauthors [2022] have used CMORPH to calculate rainfall erosivity globally, since it is available at 30-minute temporal resolution. Both IMERG and CMORPH use passive microwave observations which is spatially propagated before infra-red precipitation observations are incorporated. Both algorithms use a variety of methods to interpolate observations, and in fact IMERG uses the CMORPH Kalman filter Lagrangian time interpolation scheme [Joyce et al. 2004, 2011]. The two products perform comparably when compared with observations [Alsumaiti et al. 2020, Llauca et al. 2021, Mekonnen et al. 2021, Montes et al. 2021, Nwachukwu et al. 2020], although several studies have suggested the version 6 of IMERG performs better than CMORPH across a diverse set of climatological regimes [Llauca et al. 2021, Mekonnen et al. 2021, Montes et al. 2021, Tang et al. 2020]. It is not my intention in this study to adjudicate which of the two satellite products performs better, but instead to demonstrate the use of IMERG data to estimate erosivity in the same manner as Bezak and coauthors [2022] did with CMORPH.

## 4.3 Erosivity in Arable Zones

The implications of these erosivity estimates for geomorphological processes vary depending on the location. In steep, mountainous regions, the dominant process of erosion is bedrock landsliding [e.g., Hovius et al. 1997, Marc et al. 2019], while the impact of overland flow and rain-splash on erosion of surficial materials contributes a smaller proportion of overall erosional fluxes. RUSLE-based approaches do not consider bedrock landsliding, and as a result these erosivity estimates are not as relevant in steep, upland areas. The extremely high erosivity values observed in, for example, Papua New Guinea and the Eastern Himalayan Syntaxis may therefore not be directly correlated with overall erosional fluxes from these regions where bedrock landsliding is more dominant [Hovius and Stark, 2006].

Soil loss by purely rain-driven processes is highly relevant in areas of agricultural cultivation, in part due to broadly lower topographic steepness but also because of the potential impact of soil degradation on agricultural productivity. As such, I have compared the 30-minute IMERG erosivity estimates with datasets on the location of cropland and pastures around the world [Ramankutty et al. 2008] to assess whether highly agriculturally productive areas experience high erosivity values, or not.

At low crop densities, there is a somewhat higher variability in erosivity values than at higher crop densities (Figure 9). This is especially pronounced in Asia and North America. At higher crop densities, erosivity values are more consistent. The mean value of erosivity remains relatively consistent across all crop densities, with the exception of Europe where at moderate-to-high crop density, erosivity declines. This suggests that the location of cropland is subject to lower rainfall erosivity in Europe, but not elsewhere. Ramankutty et al. [2008] also generated estimates of pastureland, which is also a critical area for soil loss as a result of erosion. I have compared pasture area with the 30-minute IMERG estimate of erosivity (Supplementary Figure 7). Unlike cropland, in both North America and Asia as the proportion of pastureland increases, the erosivity decreases. This suggests that pastureland may be less vulnerable to highly erosive rainfall than cropland across these two continents. Although these are very spatially coarse assessments, by cross-comparing where highly erosive rain falls with agricultural zones it is possible to highlight areas of concern for soil degradation. It is notable that the variability in erosivity is significantly greater

across continents when agricultural zones are subset (compare Figure 6 and Figure 7 for South America and Africa, for example) which likely reflects the seasonality of rainfall in agricultural areas. Highly seasonal rainfall patterns would naturally drive variability in agricultural productivity, but also erosivity; planning for growing season and erosion season must therefore be considered side-by-side. However, it is also critical to note that soil erosion is not entirely dependent on rainfall erosivity, and to fully contrast areas of concern for erosion and agricultural areas, further research will need to incorporate the other parameters in the RUSLE equation including topography, land cover, and land management practices.

While the probability density function of the 30-minute IMERG erosivity and GloREDa erosivity estimates (Figure 4) show a broadly similar 2-peak distribution (albeit with the higher erosivity peak somewhat in the GloREDa estimates offset to higher values), when I limit the data to only the areas with cropland greater than 80% (Supplementary Figure 8) or pasture greater than 80% (Supplementary Figure 9), there is a significantly larger proportion of values in the GloREDa estimates at higher erosivity values. For areas with 80% or more cropland, the lower peak in erosivity values overlaps for the IMERG and GloREDa data (although mean values are higher for GloREDa, and the modal value is found in the second peak, rather than the first as in the IMERG results), but for areas with 80% or more pastureland, the entire density function of GloREDa results is significantly higher than that of the IMERG-derived erosivity. This comparison shows that the satellite estimates of erosivity suggest that global cropland and pasture areas are subject to lower rainfall erosivity than the ground-based estimates previously indicated.

**5 Conclusions**

In this study, I have used the IMERG satellite-derived precipitation product to generate a global estimate of rainfall erosivity. I have tested three methods, including two simplifications of high-temporal frequency data, and while these do produce similar global patterns of erosivity to the 30-minute data I suggest that the estimate derived from the 30-minute data is the most appropriate global model, not only because it provides a closer approximation of ground-based values in areas where the density of ground-based gauges used by GloREDa [Panagos et al. 2017] is greatest, but also since it allows for the equations used in the standard formulation of the widely-used RUSLE equation to be fully applied without simplification. A further benefit of this approach is that it allows for rapid calculation of monthly erosivity estimates, allowing researchers and practitioners to assess the peaks and troughs in erosivity across each year, rather than a single static value.

When contrasted with ground-based estimates, the IMERG-derived erosivity estimates are more similar in Europe, where the density of gauges used to calculate the ground-based estimate is higher, but in many other areas – and in particular areas of high cropland and pasture density – the IMERG estimates show lower erosivity values. Further research is necessary to ground-truth the IMERG-based estimates of erosivity in these data-poor areas, to test whether satellite-derived erosivity can be used in place of gauges, and thus maximising the use this globally-consistent dataset for erosivity.

**Code and Data availability**

The IMERG data were provided by the NASA/Goddard Space Flight Center's GPM Team and PPS, which develop and compute IMERG version 06B as a contribution to the Global Precipitation Monitoring mission, and archived at the NASA GES DISC. All methods necessary to replicate these results can be found in the text. Datasets on global crop production and yield are available at http://www.earthstat.org/, accessed November 2nd, 2022. The 20-year estimates of erosivity derived from IMERG are available in the supplementary material. All monthly data for erosivity will be released through the NASA GES DISC and code through the NASA Github upon completion of the NASA data release process, and are available upon request from the author until that date.

**Author Contribution**

R.E. Conceptualized the study, conducted data analysis, and wrote the manuscript.

**Competing Interests**

The author declares that he has no conflict of interest.

**Acknowledgements**

Robert Emberson is supported by a NASA New Investigator Program Grant.

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

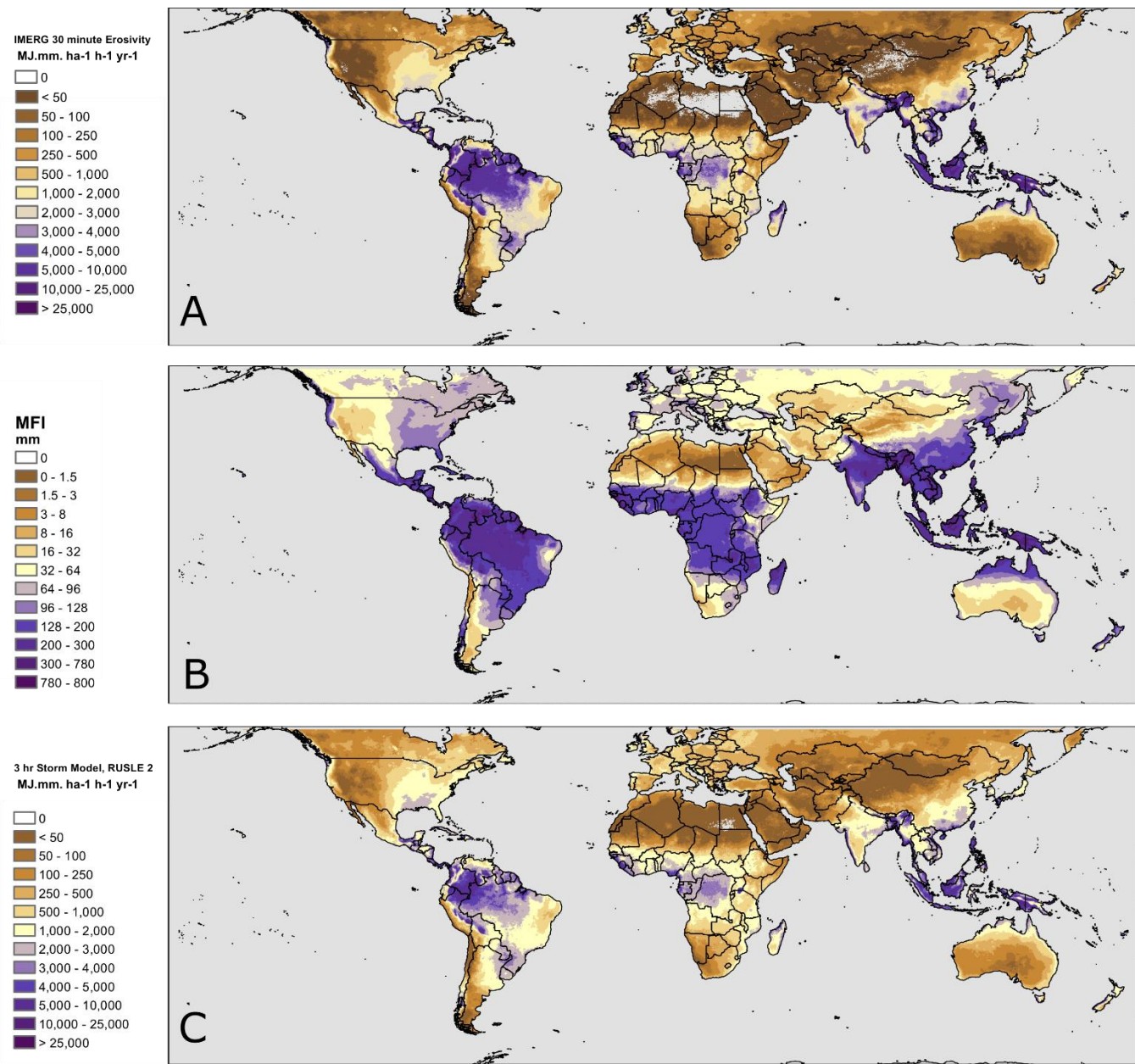

Figure 1: Global map of erosivity estimates derived from the three different approaches. (A) erosivity calculated using 30-minute IMERG data. (B) erosivity calculated using the 3-hour IMERG data and storm simplification. (C) Modified Fournier Index of erosivity. Note that the colour scheme for (A) and (B) are the same, while the MFI colour scheme has been selected to highlight the similarity in spatial patterns, rather than absolute values with the other two estimates. The overall colour scheme is selected to ensure readability to colour-blind readers.


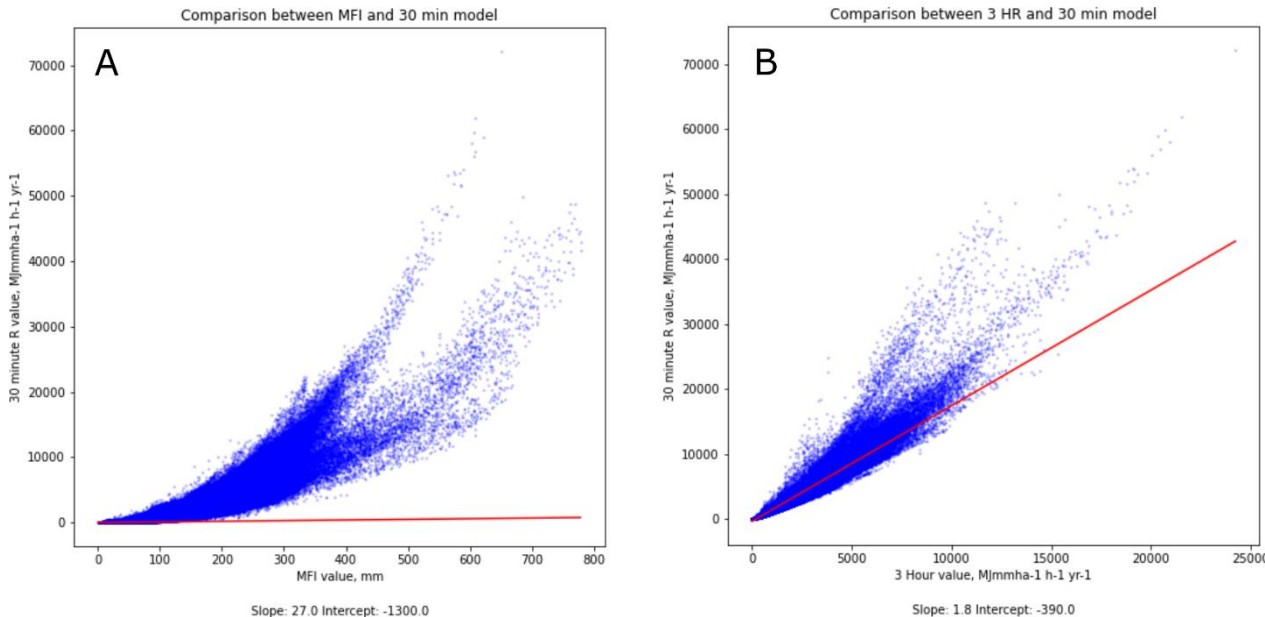

Figure 2: Comparison between cell-by-cell values of the 30-minute IMERG estimate of erosivity (y-axis) and the two other methods; (A) shows comparison between 30-minute IMERG estimate and MFI, while (B) shows the comparison between 30-minute IMERG estimate and the 3-hour IMERG data estimate. Note that the red lines in both figures show the 1:1 line, with values above that line indicating higher erosivity estimates in the 30-minute version. The 1:1 line in figure (A) does not have any physical meaning since MFI has different units, but is shown for illustrative purposes.

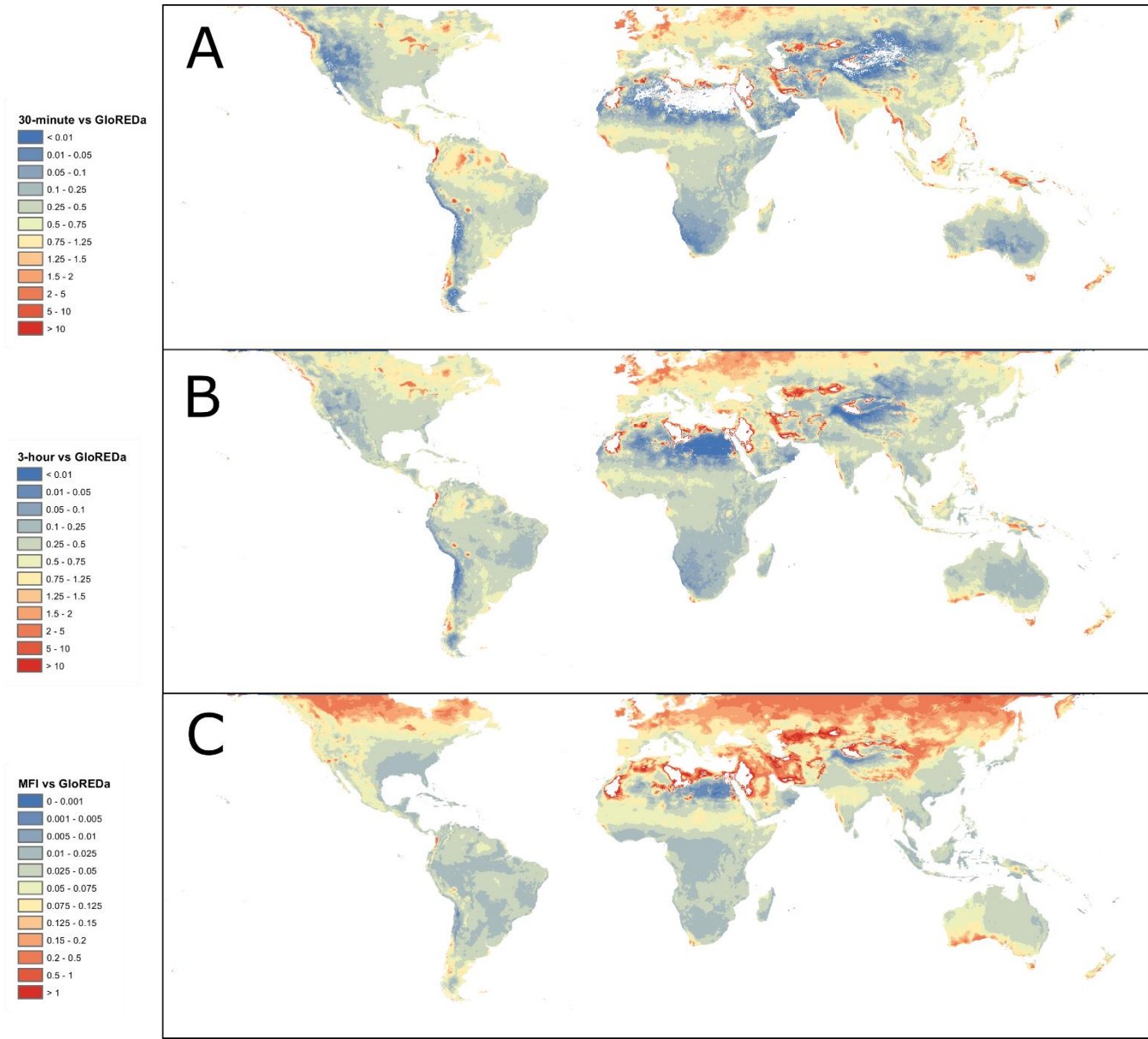

Figure 3: Global maps of the ratio between the IMERG-derived estimates of erosivity and the ground-based gauge estimates from GloREDa [Panagos et al. 2017]. The three panels show the comparison between GloREDa and (A) erosivity calculated using 30-minute IMERG data. (B) erosivity calculated using the 3-hour IMERG data and storm simplification. (C) Modified Fournier Index of erosivity. Note that the scale for (A) and (B) are identical, since they have the same units; however, the values in C are lower since the MFI has not been scaled to the other values. The scale is thus lowered to allow for comparison of spatial patterns with (A) and (B).

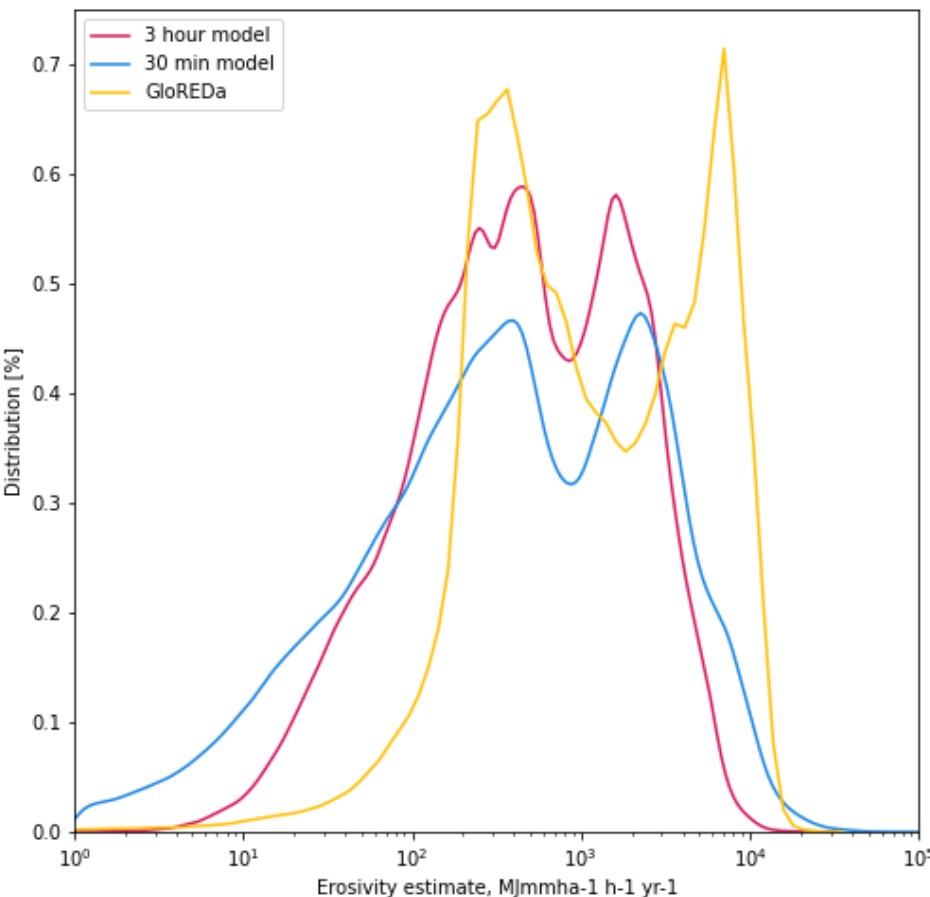

Figure 4: Probability density function for the cell-by-cell values for each of the IMERG erosivity estimates and the GloREDa estimate. MFI values are not shown since they are not the same units as the other erosivity estimates. In figure S1, the normalised values are shown, which allows comparison of the MFI values.

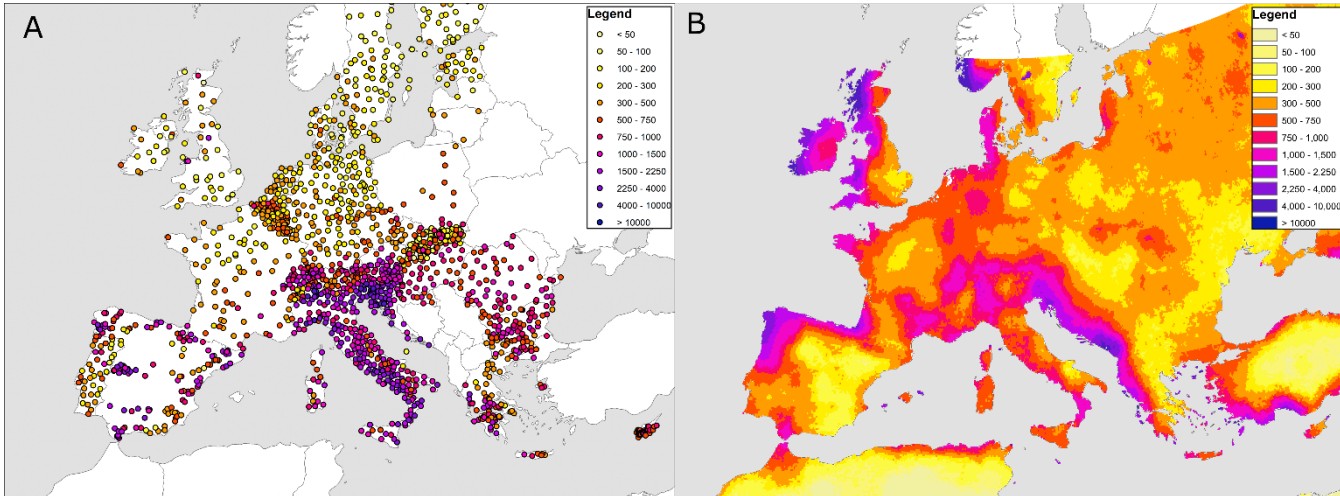

Figure 5: Comparison of gauge-derived erosivity estimates. Sub-figure A: R-values from the gauge-based REDES database (Panagos et al. 2015). Sub-figure B: R-values from the 30-minute IMERG derived estimates. The colour scheme for both datasets is the same, allowing for comparison of spatial results.

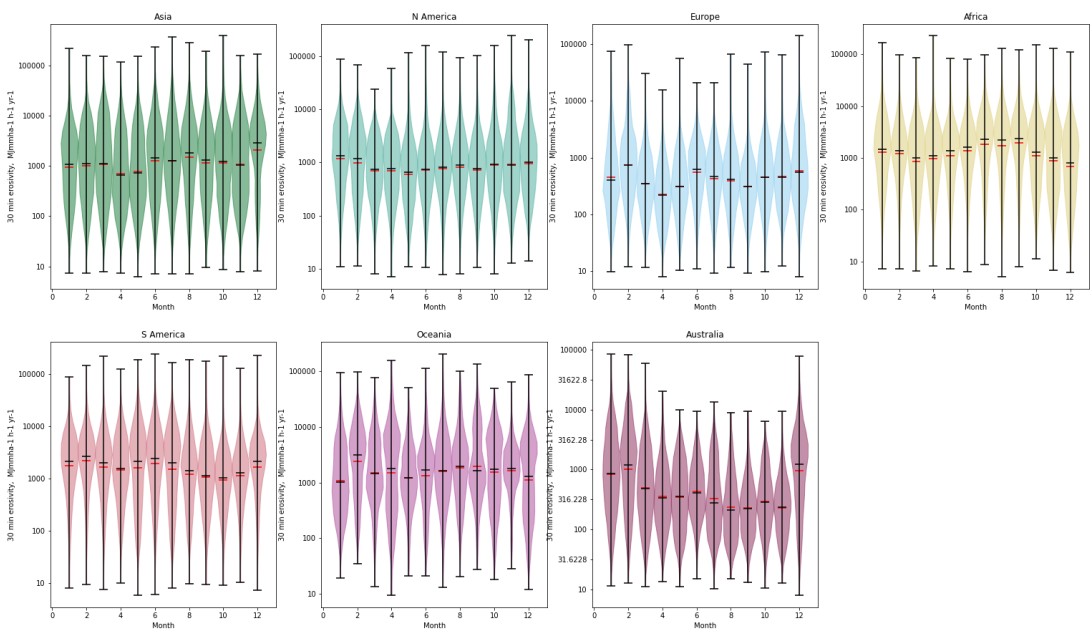

Figure 6: Monthly variability in erosivity calculated using IMERG 30-minute data, for each continent for the year 2020-2021. Median values are shown in black, and mean values with a red bar. Note that the despite the data being shown at a monthly interval, the units remain the same as for the annual erosivity, since the erosivity values are calculated for a standard annual time period, even though the data is drawn from monthly data only; this allows for consistent comparison across months of different lengths as well as with annual data.

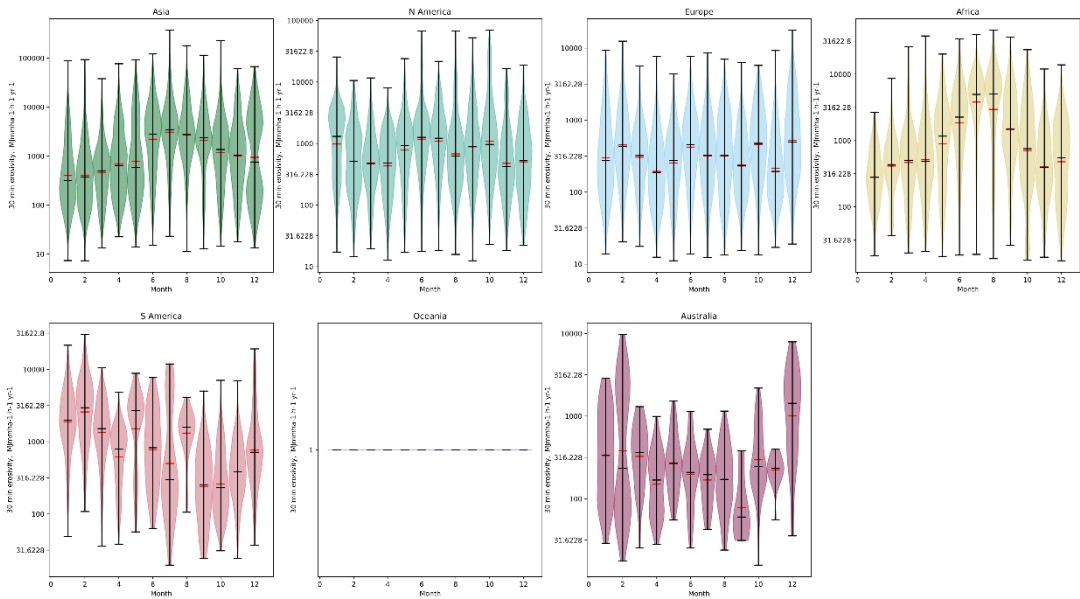

Figure 7: Monthly variability in erosivity calculated using IMERG 30-minute data, for each continent for the year 2020-2021. In this figure, only grid cells where the proportion of cropland exceeds 80% are shown. Median values are shown in black, and mean values with a red bar. Note that the despite the data being shown at a monthly interval, the units remain the same as for the annual erosivity, since the erosivity values are calculated for a standard annual time period, even though the data is drawn from monthly data only; this allows for consistent comparison across months of different lengths as well as with annual data.

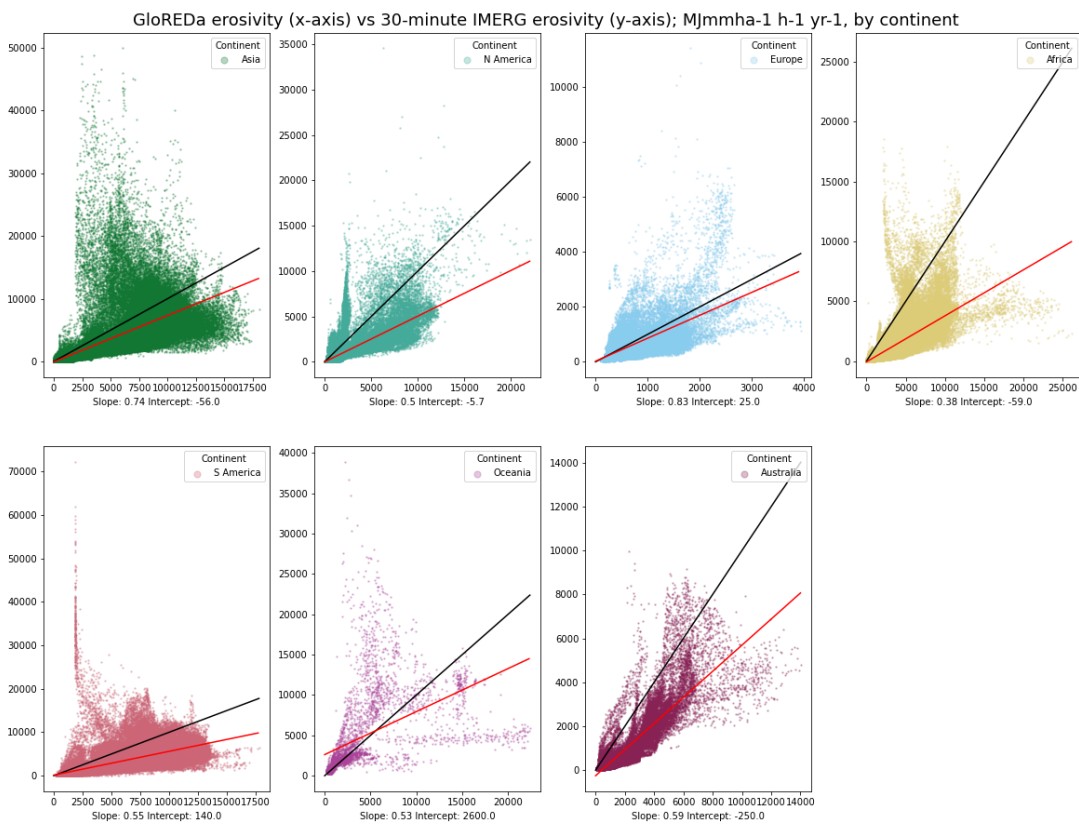

655

Figure 8: Comparison of cell-by-cell values for GloREDa (x-axis) and the 30-minute IMERG-derived erosivity estimate (y-axis) for each continent. The black line in each figure shows the 1:1 relationship, while the red line is the linear regression estimate of the best fit line for the data. Below each figure, the slope and intercept of the best-fit line are shown. All values have units of MJ.mm ha$^{-1}$ h$^{-1}$ yr$^{-1}$.

660

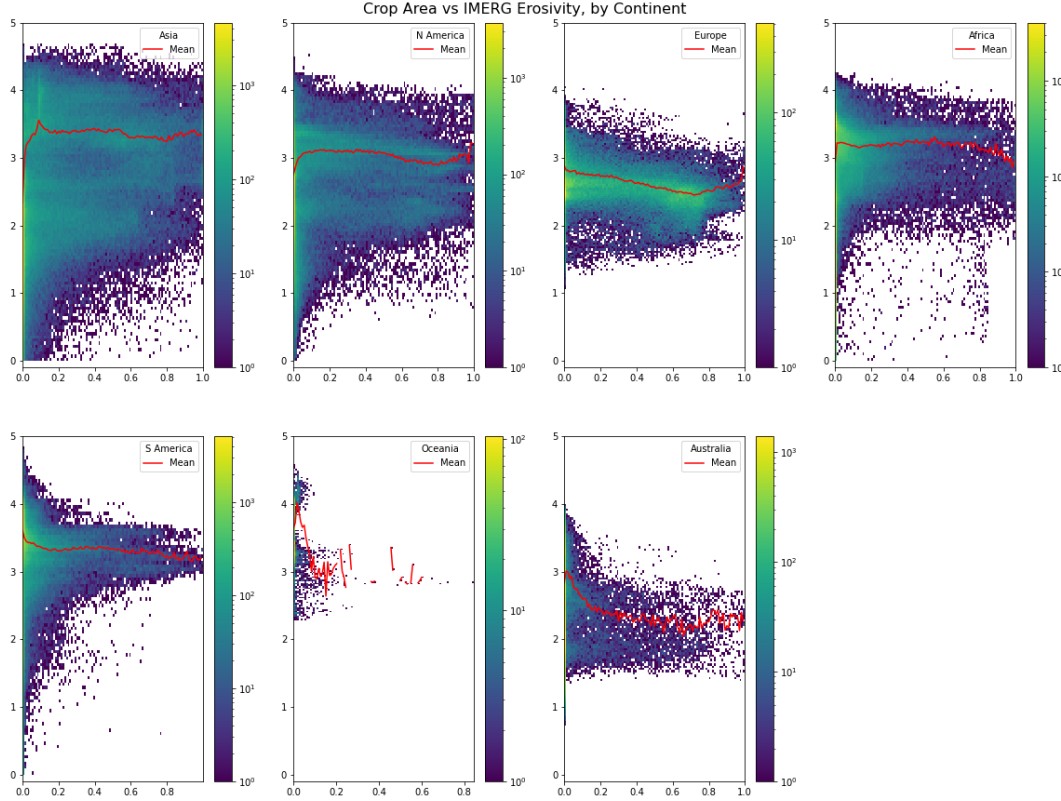

Figure 9: 2-dimensional histogram of the cell-by-cell values of the 30-minute IMERG estimate (y-axis, note logarithmic scale) and the fraction of cropland in that cell [Ramankutty et al. 2008]. The colour scale indicates the number of cells with those values. The figure shows the values distributed across each continent; continent labels are shown in the legend. The red line in each figure indicates the moving mean for different crop fractions.