# Peer review of "Dynamic Rainfall Erosivity Estimates Derived from GPM IMERG data"

_EGUsphere, 2022_

## Referee Comment (RC2)

The manuscript "Dynamic Rainfall Erosivity Estimates Derived from GPM IMERG data" evaluated the performance of IMERG-Final rainfall product in estimating global rainfall erosivity estimates. Three methods were tested in this manuscript.

The main drawback of the manuscript is that it lacks high quality Introduction, and explanations that describe the reasons for getting those results. The author also uses many subjective words, which is not recommended for scientific writing. Specific values are encouraged.

Specific comments:

1. Lines 16-18: I think the differences between the IMERG-derived and gauge-derived estimates cannot suggest that the IMERG data may allow for improved erosivity estimates in ungauged areas unless the author can provide corresponding evidence. The findings should be supported by study results. I noted that the IMERG-derived erosivity estimates are more similar to ground-based estimates in areas with a high spatial density of gauges, such as Europe. This means that the quality of erosivity estimates based on IMERG depends on the spatial density of gauges if the ground-based estimates are regarded as "true". So, the erosivity estimates based on IMERG might have large errors and uncertainties in ungauged areas due to error propagation.

2. Line 33: please give specific factors, and add citations.

3. I believe that the Introduction section needs to be rewritten because of the lack of many previous studies, although the author introduced two papers that focused on

estimating soil erosivity using CMORPH and TMPA, respectively. In fact, there are

many papers investigating soil erosivity using multiple gridded precipitation

datasets. Given that this study aims to evaluate the potentials of the IMERG data

and different methods in estimating erosivity, by analyzing the advantages and

limitations of using different rainfall datasets and different approaches in estimating

erosivity, it can draw out the reason why you did this study and what scientific

questions this study will solve. So, please review related papers and improve the

Introduction section.

4. Lines 49-50: please give the specific methods directly, rather than vague

   descriptions.

5. Lines 65-66: IMERG-Final is derived from the error correction of IMERG-Late by

   using GPCC as a reference.

6. Lines 70-74: how to separate or remove solid precipitation from precipitation? I

   suggest the author describe this point as much as possible because errors may be

   introduced in erosivity estimates when the solid precipitation cannot be removed

   accurately.

7. Lines 120:please provide citations for this sentence.

8. Eq.7: what is $I30_k$ and k, please provide specific meaning.

9. Figure 1 A: no colour bar to represent the 0 value.

10. Line 179: Figure 1C?

11. Lines 182-183: which papers supported this point?

12. Lines 188-190: why do the differences increase with increasing erosivity, and why

does the 30-minute model generally produce higher values? I think the readers will be interested in the reasons. In addition, underlying reasons might be helpful for us to find a more reliable method to accurately estimate erosivity.

13. Lines 200-207: lack of explanations for the results.

14. Lines 229-230: how did you judge the absolute estimates from the 30-minute model are closer to those of GloREDa? Based on the results in Table 2?

15. Captions of Figures 5 and 6: 2020 or 2021?

16. It is an interesting phenomenon that the IMERG estimates are much larger than those of GloREDa in several coastal areas, why does this phenomenon appear? Is it caused by large errors in IMERG precipitation estimates or other reasons?

17. I suggest the author discuss potential reasons for large differences between IMERG-derived estimates and GloREDa estimates.

18. Line 334: delete "is".

19. The author claimed that "These estimates provide informative comparisons to the study of Bezak and coauthors [2022]" (in line 49). However, I did not see the relevant comparison test in this study. In addition, the comparison is possible to demonstrate that IMERG performed better than CMORPH in estimating erosivity. So, a comparison may be added to demonstrate that IMERG is more suitable for use in estimating erosivity compared to CMORPH. More importantly, I think the erosivity obtained from this work is not limited to comparison with Bezak and coauthors [2022], please use more appropriate sentences to sublimate your work.

---

## Author Comment (AC2)

Response to Reviewers: Emberson, 2023 (EGUSPHERE)

I am grateful to the editor for assembling three constructive and informative reviews of the submitted study. Each of the reviewers and the additional commenter have provided exceptionally helpful points that I believe will serve to improve the study at hand. Based on these reviews, I feel that I will be able to revise the study to be a significant improvement over the initially submitted study. In the response below, I have provided replies to each of the reviewers' comments, including how I intend to revise the study. The responses are provided in red text.

Reviewer 1:

In the submitted paper author investigates the performance of the GPM IMERG rainfall data to calculate global rainfall erosivity. Three approaches are tested for the estimation of the global rainfall erosivity patterns. A detailed comparison with the GloREDa dataset and derived global erosivity map is conducted. The submitted paper is in the scope of the HESS journal. It is very well written and it is easy to follow the conducted methodological steps and discussion of the results. The presented results can be regarded as an important step towards determination of the so-called dynamic global rainfall erosivity maps that could be used as input data for the soil erosion models. Therefore, I only have a few moderate comments/suggestions:

I am grateful to the reviewer for their time to review this study and their thoughtful comments. I appreciate the support for the study.

Can author add some additional (besides what is written in sections 4.2 and 4.1) discussion about the uncertainty related to the satellite-based rainfall dataset and how does this transforms into the rainfall erosivity estimation and does uncertainty perhaps has some seasonal patterns. Would it be perhaps possible to compare calculated IMERG monthly rainfall erosivity to REDES dataset (subset of GloREDa data; monthly rainfall erosivity maps for Europe).

The reviewer raises an important point about the uncertainty associated with the IMERG rainfall data. The geographical uncertainty associated with satellite rainfall data has been discussed in detail by Tian and Peters-Lidard (2010). With higher rainfall rates, the uncertainty values tend to be lower, with larger uncertainty over higher latitudes during the colder season between different rainfall products. High relative uncertainties also exist across complex terrain like mountain ranges and coastlines. While there are not published studies to my knowledge that have assessed the relative uncertainty associated with IMERG version 6B, the same characteristics are likely true – higher degree of certainty across inland, flatter areas, with greater reliability at higher rainfall rates. This is critical since it describes much of the worlds agricultural areas. I intend to add a description of satellite rainfall uncertainty to the discussion section.

In addition, the reviewer raises the important question about comparison with the REDES dataset. This is also related to the question posed by reviewer 3 regarding comparison to ground based data. I have conducted a comparison of the gauge-based estimates from the REDES dataset with the R-factor

estimated from the IMERG data. The REDES data covers only Europe, which is a major limitation of this comparison. However, this is also to my knowledge the largest dataset of rainfall erosivity values calculated from gauge station values openly available.

I have compared the long-term calculated R-values from the REDES data with the IMERG-derived estimates for the pixel in which the REDES data is drawn. Although this dataset only covers Europe, it is an extensive dataset and provides informative comparisons. Spatially, the IMERG based analysis performs well in several European countries, including Greece, the Iberian peninsula, Germany, France, Switzerland and the low countries. IMERG overestimates in comparison with gauges in the Western parts of the United Kingdom and Ireland, while somewhat underestimating in Italy and parts of Bulgaria and Romania. Given that the gauges that make up REDES are not uniformly distributed, a statistical comparison of the two datasets will be dominated by countries with higher gauge density (like Belgium, Italy, and Slovakia), so a spatial comparison is more informative, although I have additionally calculated the statistical correlation between the gauge-estimated R-values and IMERG-estimated R-values. I intend to include this analysis and results in a revised manuscript.

At the end of the manuscript, author wrote that more detailed comparison is needed with ground-based data in order to verify the IMERG dataset. Could perhaps some additional discussion be added regarding this possible further step (e.g., how this could be done). It should be also noted that also GloREDa dataset has its own limitations (e.g., different data periods were used for different stations, data temporal resolution was not uniform, etc.). Even if assume that GloREDa represents the "true" rainfall erosivity patterns it is question, how accurate this actually is. Is there any alternative way, without using GloREDa dataset. Perhaps satellite-based drop-size-distribution data and calculation of the rainfall erosivity directly from the DSD?

This is an excellent point. As discussed in the response to reviewer 3 below, and in the discussion above relating to REDES, I will endeavor to use gauge-based estimates as a more direct comparison with the IMERG estimates. This is not a globally comprehensive analysis – as the commenter on the article notes, only the European portion of the global gauge data used to calibrate GloREDa is currently publicly released, so I am limited to that comparison at present. In revising this manuscript, I will provide an explanation of the use of the REDES data for comparison, as well as areas where further research would be critical – this particularly includes areas with minimal gauging where there is significant divergence between the satellite and gauge-based estimates, such as coastal areas and deserts.

Figure 2: It would be perhaps good to include the 1:1 line to the first (A) figure as well.

This is a good point – I will add this to a revised version of the paper. It is naturally important to note that the 1:1 line between MFI and the 30-minute R factor estimate does not reflect any physical connection. I will add text to the caption to explain that the 1:1 line is merely illustrative.

Figure 4: Units should be added for x-axis. Is it meaningful to include MFI data? I see that you included

Figure S1 but at least it should be noted in the Figure 4 caption that that MFI has different units. Perhaps the same could be done for Figure S7 and Figure S8.

Thanks to the reviewer for raising this point. I am inclined to agree that the figure without units is not comprehensive. Since the MFI values are not the same unit, I am reluctant to include this part in Figure 4. For the relative scale in Figure S1, I think it is sensible to keep the MFI values for illustrative purposes. I will amend figure 4 to remove the MFI values and add units, as well as amend the caption to reflect that.

Figure 5: Perhaps the y-axis caption is not the most clear, it should be total rainfall erosivity in specific month (units mo-1), right? Also it would be perhaps better to plot the absolute values (not log) since this can be easier to put in context (compared to annual erosivity values). Additionally, figure caption should specify what do black and red lines represent (mean and median?).

Figure 6: Similarly, as for Figure 5.

Thank you for raising this point. Median values are shown in black, and mean values with a red bar. Note that the despite the data being shown at a monthly interval, the units remain the same as for the annual erosivity, since the erosivity values are calculated for a standard annual time period, even though the data is drawn from monthly data only; this allows for consistent comparison across months of different lengths as well as with annual data. I will add this explanatory text to the final version of the text.

For the values – I will use different axis labels to ensure that the scales are appropriately comparable with the annual data. Thank for this suggestion.

Figure 6: It is interesting why the monthly variations for S. America (and also for some other continents) are much larger as compared to Figure 5 (this is noted also in the text), any explanation?

This is a good point. This likely reflects the seasonality of rainfall in agricultural areas. Highly seasonal rainfall patterns would naturally drive variability in agricultural productivity, but also erosivity; planning for growing season and erosion season must therefore be considered side-by-side. I will include text in a revised version in section 4.3 to explain this.

Reviewer 2:

The manuscript "Dynamic Rainfall Erosivity Estimates Derived from GPM IMERG data" evaluated the performance of IMERG-Final rainfall product in estimating global rainfall erosivity estimates. Three methods were tested in this manuscript.

The main drawback of the manuscript is that it lacks high quality Introduction, and explanations that describe the reasons for getting those results. The author also uses many subjective words, which is not recommended for scientific writing. Specific values are encouraged.

I thank the reviewer for taking the time to review the manuscript and providing thoughtful and constructive feedback.

Specific comments:

1. Lines 16-18: I think the differences between the IMERG-derived and gauge-derived estimates cannot suggest that the IMERG data may allow for improved erosivity estimates in ungauged areas unless the author can provide corresponding evidence. The findings should be supported by study results. I noted that the IMERG-derived erosivity estimates are more similar to ground-based estimates in areas with a high spatial density of gauges, such as Europe. This means that the quality of erosivity estimates based on IMERG depends on the spatial density of gauges if the ground-based estimates are regarded as "true". So, the erosivity estimates based on IMERG might have large errors and uncertainties in ungauged areas due to error propagation.

   The reviewer raises a really good point. To clarify, the IMERG rainfall estimates use gauges around the world as a way to anchor the satellite observations to ground values, but the degree of weighting for the correction is lower in areas with lower gauge density, so that the IMERG values are less influenced by individual gauge peculiarities. In other words, where gauges are less dense, IMERG is more independent from the gauges.

   This is important, since the GloREDa estimates are interpolated purely from gauges. Where gauge density is low, GloREDa is dependent to a larger degree on individual gauges – which is not the same as IMERG, as mentioned above. This means in areas with fewer gauges, there is likely to be a greater divergence between the two estimates.

   The more important point here is the question about what is a 'true' value for erosivity. It is clear that the IMERG-derived estimates are based on large scale satellite observations, which have their own systematic errors (see further comments below), and as such should not be considered to be a true value for erosivity. The gauge based estimates, such as REDES (Panagos et al. 2017) are true in a specific location – where the gauge was sited. All three reviewers have highlighted that it is important to compare more directly to the gauge based estimates, and as such I intend to make this a central point of the discussion in a revised manuscript.

   However, the larger intent of this study is to provide a global-scale estimate of erosivity. As such, I suggest it is important to compare the results of the satellite-derived global estimate (here) and

the gauge-derived global estimate (GloREDa). Where the gauge density is high, GloREDa is likely to outperform IMERG (and after comparing the two estimates with the REDES gauge dataset from Europe, this is true – although since GloREDa is directly calibrated by REDES, this is perhaps to be expected). However, in areas where gauge density is low, such as Africa and South America, IMERG is less dependent on specific gauges.

The reviewer is correct that I cannot explicitly say that IMERG outperforms the gauge-based estimates in those areas **because there are no gauges to provide true R-values**, and as such in revising this study I will ensure that any specific mention of 'improved performance' will be edited. However – as mentioned in the initial study, there are locations with no gauges, such as the Atacama and Namib deserts, where GloREDa predicts physically impossible values for erosivity due to interpolation between limited gauges since there is negligible rainfall in these locations, and IMERG provides a much lower estimate. I suggest that I can therefore argue that IMERG *may* be a more useful tool in areas of lower gauge density for calculating erosivity.

2. Line 33: please give specific factors, and add citations.

    I will add the following text to elaborate: Degradation of soil is driven by many factors including cover by impermeable materials, physical compaction, wind and rain-driven erosion, salinization and chemical degradation (Ferreria et al. 2018 ).

3. I believe that the Introduction section needs to be rewritten because of the lack of many previous studies, although the author introduced two papers that focused on estimating soil erosivity using CMORPH and TMPA, respectively. In fact, there are many papers investigating soil erosivity using multiple gridded precipitation datasets. Given that this study aims to evaluate the potentials of the IMERG data and different methods in estimating erosivity, by analyzing the advantages and limitations of using different rainfall datasets and different approaches in estimating erosivity, it can draw out the reason why you did this study and what scientific questions this study will solve. So, please review related papers and improve the Introduction section.

    I appreciate the comment from the reviewer here. While the intent of this study has been to focus on other satellite-derived rainfall datasets used for erosivity, the reviewer is correct that other gridded datasets have been employed, including Borelli et al. (2016), Yin et al. (2017), Raj et al. (2022) and Chen et al. (2022). I will include a discussion of these studies in a revised introduction section.

4. Lines 49-50: please give the specific methods directly, rather than vague descriptions.

    While I appreciate the point the reviewer makes, I respectfully disagree. I am of the belief that the methodology section is the place to provide specific details about methodology rather than the introduction. I suggest that the purpose of the final paragraph of the introduction is to provide a high-level overview of the study.

5. Lines 65-66: IMERG-Final is derived from the error correction of IMERG-Late by using GPCC as a reference.

I appreciate the reviewer catching this mistake. I will correct the mention of IMERG-Early to IMERG-Late in a revised manuscript.

6. Lines 70-74: how to separate or remove solid precipitation from precipitation? I suggest the author describe this point as much as possible because errors may be introduced in erosivity estimates when the solid precipitation cannot be removed accurately.

   I appreciate the reviewer raising this point as it allows me to clarify the methodology. IMERG-Final data includes a specific field that estimates the fraction of liquid precipitation in each grid cell for each data point. The liquid precipitation was therefore calculated by multiplying the total precipitation field by the fraction of liquid precipitation field. I will add this specific description to a revised manuscript.

7. Lines 120:please provide citations for this sentence.

   Thank for catching this – I will add the citations of Arnoldus (1977) and Renard and Freimund (1994)

8. 7: what is and k, please provide specific meaning.

   Thank you for flagging this – k here represents the kth storm (i.e., each storm from 1-k). I will add this to the revised text.

9. Figure 1 A: no colour bar to represent the 0 value.

   Thank you for raising this – I will add this in revision.

10. Line 179: Figure 1C?

    Thank you for catching that – I will correct it in a revised paper.

11. Lines 182-183: which papers supported this point?

    I will add a citation here, citing Renard and Freimund (1994) and Smithen and Schulze (1982)

12. Lines 188-190: why do the differences increase with increasing erosivity, and why does the 30-minute model generally produce higher values? I think the readers will be interested in the reasons. In addition, underlying reasons might be helpful for us to find a more reliable method to accurately estimate erosivity.

    This is a good point, and I appreciate the reviewer flagging it so that I can explain to readers. I will add the following text to the manuscript:

    Although the 3-hour rainfall approach includes all 3-hour windows, rather than excluding smaller rainfall events (as described in the methodology above), the 30-minute approach captures rainfall events with larger short-term rainfall intensity, which will result in larger erosivity estimates since the scaling between rainfall intensity and erosivity is non-linear.

13. Lines 200-207: lack of explanations for the results.

Thank you for raising this point. In a revised manuscript, I will provide a significantly larger discussion of the reasons why GloREDa and IMERG diverge. In particular, in the IMERG limitations section, I will discuss new analysis:

It is notable that the IMERG-derived results are lower than the gauge-derived results in a number of locations and are lower on average in all continents except Oceania (Table 2), and so underestimation of rainfall events driving erosivity may be a reason for this. Bezak et al. (2022 highlighted that the largest 11% of rainfall events contribute 50% of the erosivity, so it is particularly relevant not to miss these very large events. Since IMERG may miss very short-lived rainfall events, it is especially important if rainfall driving erosion is from extremely short-lived rainfall events, rather than longer storms. To explore this, I have analysed storm histories from four locations around the world; two in areas of concern for soil erosion (Near Wichita, USA, and Lucknow, North India) one in a critical region of degradation where the IMERG estimate exceeds GloREDa (Central Sierra Leone) and near San Pedro de Atacama, in the dry desert of Northern Chile. In Chile, only 3 rainfall events are observed over the entire 2000-2021 interval. In the other locations, I tested what proportion of the storm events in each location is formed by the 30-minute period of rainfall and compared that to the total storm rainfall. Storms where the most intense short bursts of rainfall make up most of the total rainfall are likely to be more underestimated by IMERG in comparison with storms where more consistent rainfall is observed. In Supplementary Figures 5A-D, I show the fraction of total rainfall in each storm from the 30-minute peak rainfall vs the cumulative kinetic energy from rainfall. In Sierra Leone and Lucknow, more than 80% of rainfall kinetic energy is derived from storms where the 30-minute interval of peak intensity is less than 50% of total rainfall. In Wichita, the storms are more dominated by the short term intense rainfall – 80% of kinetic energy is derived from storms where the maximum 30-minute rainfall forms up to 80% of total storm rainfall. I suggest that in locations like Wichita, IMERG may be more subject to missing short-term bursts of rainfall. This may explain why IMERG is lower than GloREDa in Wichita and the US South East.

14. Lines 229-230: how did you judge the absolute estimates from the 30-minute model are closer to those of GloREDa? Based on the results in Table 2?

I appreciate the reviewer raising this point. I will remove this text, since 'closer' implies a more singular answer than is possible. In particular, the probability density estimates (Figure 4) are the best way to illustrate the differences between GloREDa and IMERG results, and these are already discussed elsewhere in the text.

15. Captions of Figures 5 and 6: 2020 or 2021?

Thank you for flagging – I will amend to 2020-2021.

16. It is an interesting phenomenon that the IMERG estimates are much larger than those of GloREDa in several coastal areas, why does this phenomenon appear? Is it caused by large errors in IMERG precipitation estimates or other reasons?

In a revised manuscript, I will provide additional context for this point. In particular, this will draw on new comparison with the REDES gauge data:

Since GloREDa is an interpolated dataset based on gauge-derived estimates of erosivity, the specific values in a given grid cell will not always represent the exact gauge-derived value for a given pixel. To account for this, and to compare the IMERG-derived values with those of ground-based gauges used to calibrate GloREDa. The Rainfall Erosivity Database on the European Scale (REDES, Panagos et al. 2015 ) is an openly available dataset of gauge-derived estimates of erosivity. In Figure 5, the values from the IMERG-based estimates for Europe are shown in comparison with the values from the REDES dataset. Spatially, the IMERG based analysis performs well in several European countries, including Greece, the Iberian peninsula, Germany, France, Switzerland and the low countries. IMERG overestimates in comparison with gauges in the Western parts of the United Kingdom and Ireland, while underestimating in Italy and parts of Bulgaria and Romania. Given that the gauges that make up REDES are not uniformly distributed, a statistical comparison of the two datasets will be dominated by countries with higher gauge density (like Belgium, Italy, and Slovakia). The slope of the relationship between the two datasets is 0.26, while the slope of the relationship with GloREDa is 0.5 – although given that this dataset is used to calibrate GloREDa, this is not unexpected.

New Figure 5:

[Figure]

Figure 5: Comparison of gauge-derived erosivity estimates from the REDES database (Panagos et al. 2015) and the 30-minute IMERG derived estimates. The colour scheme for both datasets is the same, allowing for comparison of spatial results.

As well as continent-by-continent differences, there are other clear zones of divergence between the IMERG and GloREDa estimates. As mentioned above, these include several coastal areas in India, Southeast Asia, and the Pacific coastlines of Colombia and Canada, where the IMERG estimates exceed those of GloREDa by a significant degree [Figure 3]. These areas are all areas where both IMERG (Supplementary Figure 4) and GPCC-gauged rainfall is extremely high [Schneider et al. 2014], and the IMERG-derived erosivity estimates broadly mirror the spatial patterns observed in the annual gauged rainfall totals.

17. I suggest the author discuss potential reasons for large differences between IMERG-derived estimates and GloREDa estimates.

I appreciate the reviewer raising this point. I suggest that the responses to points 13 and 16 (above) provide more detail and specific reasons for the differences between IMERG and GloREDa.

18. Line 334: delete "is".

Thank you for catching that error – I will remove 'is' in a revised manuscript.

19. The author claimed that "These estimates provide informative comparisons to the study of Bezak and coauthors [2022]" (in line 49). However, I did not see the relevant comparison test in this study. In addition, the comparison is possible to demonstrate that IMERG performed better than CMORPH in estimating erosivity. So, a comparison may be added to demonstrate that IMERG is more suitable for use in estimating erosivity compared to CMORPH. More importantly, I think the erosivity obtained from this work is not limited to comparison with Bezak and coauthors [2022], please use more appropriate sentences to sublimate your work.

I thank the reviewer for this point, since it certainly helps tighten the text. The reviewer is correct that I do not explicitly compare these results with the CMORPH derived results of Bezak et al. (2022). Ultimately, this is because the final erosivity results of Bezak et al. (2022) are not available (the reader must derive them themselves), so a comparison is not within the scope of this work. As a result, I will remove this sentence from the final manuscript.

Reviewer 3:

In this study, the author investigates the potential of the "Integrated Multi-satellitE Retrievals for GPM (IMERG) data" to estimate rainfall erosivity at almost global scale. Multiple approaches were tested, and the results were compared against a global map of rainfall erosivity obtained through a Gaussian regression model.

I believe that the subject of the study is interesting and within the scope of Hydrology and Earth System Sciences. I do not have a large number of comments to make. The methodology to estimate rainfall erosivity is *per se* quite straightforward. In addition, several studies applied and discussed rainfall erosivity estimates, including satellite data as rainfall input data. And last but not least, the author made a good job in the application and presentation of the results obtained.

I have the feeling that the author aimed at keeping the study concise. Which is fine, despite some more detail in the Introduction and in the Discussion sectors could make the study more comprehensive. The description of the results is good but their discussion and implications could be improved. Also the abstract could be enriched by presenting some results of the statistical analysis carried out to support the statement made.

However, in my amble opinion, the *conditio sine qua non* to suggest publication rest on the data evaluation. In its current form, this study do not provide a valid evaluation of its results. The author limits the validation exercise by comparing the multiple results obtained against the GloREDa map. However, GloREDa is fruit of an interpolation, and therefore an estimate as well. To meaningfully evaluate the performance of the approach described in this study, the rainfall erosivity estimates should be evaluated against a set of true data, or, better said, rainfall erosivity values (annual average or single storms) estimated in several meteorological stations. Adopting a statistical significant number of meteorological stations around the globe and across different climate zones. As it was done to evaluate GloREDa and other studies using satellite data at global (Bezak et al. 2022; Liu et al. 2020) or regional scale (Kim et al. 2020).

Without a proper evaluation/validation of the results against true data, in my opinion the study cannot be published. The comparison against GloREDa is a good exercise but cannot assess the performance of the estimates provided in this study. Therefore, I would recommend to the author to evaluation/validation his results against a dataset of measured rainfall erosivity data. Once this is done, and the results fully evaluated and described in the paper, the paper will make a fine contribution to current literature.

I appreciate the work of the reviewer to dig into the manuscript and provide commentary. Given that the reviewer does not provide specific line-by-line commentary, I am glad to see that they feel the bulk of the text remains sound.

The specific, singular comment from the reviewer is that there is no validation of the results with respect to ground-based data, focusing rather on the comparison with GloREDa. The reviewer is correct in this sense, and although in the initial version of this study I felt it was outside of the scope of the manuscript and work to conduct ground-based validation, I am inclined to agree with the reviewer that ground-based comparison with improve the overall impact of the study.

As mentioned in Panagos et al. (2017) although there are a large range of gauges around the world, only a small proportion have the requisite data frequency to estimate R values accurately. As such, I have obtained access to the REDES dataset developed for Europe by Panagos et al. (2015) that was also used by Panagos et al. (2017) and Bezak et al. (2022), and I suggest that this will address the major concern of the reviewer here.

As mentioned in the response to reviewer #1, I have compared the long-term calculated R-values from the REDES data with the IMERG-derived estimates for the pixel in which the REDES data is drawn. Spatially, the IMERG based analysis performs well in several European countries, including Greece, the Iberian Peninsula, Germany, France, Switzerland and the low countries. IMERG overestimates in comparison with gauges in the Western parts of the United Kingdom and Ireland, while somewhat underestimating in Italy and parts of Bulgaria and Romania. Given that the gauges that make up REDES are not uniformly distributed, a statistical comparison of the two datasets will be dominated by countries with higher gauge density (like Belgium, Italy, and Slovakia), so a spatial comparison is more informative, although I have additionally calculated the statistical correlation between the gauge-estimated R-values and IMERG-estimated R-values. The slope of the relationship between the two datasets is 0.26, while the slope of the relationship with GloREDa is 0.5 – although given that this dataset is used to calibrate GloREDa, this is not unexpected. The IMERG results are significantly below the gauged results across much of Italy, which has a very high number of gauges represented in the REDES dataset. IMERG and other satellite rainfall datasets have lower accuracy in topographically complex settings, and worse performance of IMERG in capturing intense rainfall in the mountainous parts of Italy may be one of the reasons for the underestimation of the IMERG-based erosivity estimates, although further research and comparison with ground-based gauges is certainly warranted. I intend to include this analysis and results in a revised manuscript.

Additional Commenter:

A very interesting article which focus on a topic getting a lot of attention. Obviously, the erosivity Is better estimated with high temporal rainfall data.

I appreciate the time taken by the reader to comment on the article – this is additional input that is certainly welcome.

According to our experience, the satellite products are not yet "mature" enough to capture the variability of rainfall erosivity. According to Bezak (2021), 11% of the erosive events contribute to around 50% of the total erosivity. This was done based on the detailed rainfall erosivity records in GloREDa.

Similar observations have been done by Matthews et al. (2022) based on 300,000 erosive events.

Therefore, based on measured erosivity data, very few events are the ones who contribute to major part of total erosivity. Unfortunately, the satellite products are not yet mature enough to capture the high erosive events.

The satellite products are tending to smooth the high erosive values.

This is overall an excellent point. I want to provide a full answer, but first I want to stress that I think the value of satellite-based and gauge-based data is very different, and any discussion of the two side-by-side does need to acknowledge their relative merits.

First – I am not sure I completely agree that satellite rainfall is guaranteed to underperform gauges when it comes to estimating very large rainfall events. For areas where gauge density is very high, this is likely to be true, but the local rainfall intensity in extreme rainfall events can vary widely and associated wind can also interrupt effective gauge measurement (e.g. Medlin et al. 2007). Satellite rainfall can underestimate extreme rainfall if the overpass of the Microwave satellite does not coincide with peak rainfall intensity, but studies have also shown that the uncertainty of satellite rainfall products is lower at higher rainfall intensities (e.g. Tian and Peters-Lidard (2010)).

Using Bezak et al. 2021 as inspiration, I have explored the storm histories of 4 locations from around the world; two in areas of concern for soil erosion (Near Wichita, USA, and Lucknow, North India) one in a critical region of degradation where the IMERG estimate exceeds GloREDa (Central Sierra Leone) and also in the Atacama. Bezak et al. (2021) clearly show that the 11% of erosive events are responsible for the bulk of erosivity, but I wanted to test what maximum rainfall values would correspond to the storms most responsible. In Lucknow and Wichita, more than 80% of the total rainfall kinetic energy calculated from the satellite data comes from storms with a peak rainfall intensity of <30mm/hr. In Sierra Leone, this value is much higher (<70mm/hr), but the same overall pattern emerges. In the Atacama, only three rainfall storm events were found over 20 years of data.

Although one could argue that the satellite products may just be missing the large storms, the fact that the Sierra Leone case shows the same patterns as in Lucknow and Wichita suggests that the satellite is capable of capturing rainfall events far in excess of the 30mm/hr events (or lower) that are relevant in those locations (since it does so in Sierra Leone). I suggest that although the larger rainfall events do contribute the bulk of the erosivity, those rainfall events are in many locations are not reaching a maximum intensity that would lead them to be significantly underestimated; the key types of rainfall event that are underestimated by satellite products are major tropical storms (Marc et al. 2022),

occurring in areas of extreme topography, where rainfall may exceed 100-200mm/hr – somewhat less relevant to agricultural soil degradation.

Overall – I agree that there are challenges to using satellite precipitation data, and in revising this study I want to avoid giving the sense that one product is 'better' than the other. It is clear that GloREDa is highly valuable in large parts of the globe. In revising this study I hope to provide a more nuanced perspective about the value of satellite vs gauge based products, but not to suggest that one or the other is superior.

Therefore, there will be differences between your results and the ones of GloREDa in high erosive areas (continents). That is the case for Table 2.

I would also to propose a comparison per climatic zone where you will find big discrepancies in the tropical zones.

This is certainly interesting, and something I will be pursuing in future study.

The global erosivity map (Panagos et al., 2017) has been tested and evaluated against the 3,625 measured R-factor values. Please see the figure 4 in the Panagos et al. (2017) and the excellent performance of Global assessment.

As you state in the manuscript, seems that you have estimated better than GloREDa which is not the case. If you insist your statement, you should prove that you perform better than GloREDa in the measured 3,625 stations. The GloREDa measured stations data will be available soon with a data paper.

I look forward to seeing the data! Thank you for letting me know. In revision, as mentioned above, I want to rephrase to ensure it is clear I am not stating that the satellite product is 'better' that GloREDa – this is more a comparison of the two for illustrative purposes.

Another remark: The MFI is much problematic and this has been shown in a recent review of Chen et al (2023) .

In this case, I entirely concur. In revising the study, I want to ensure that it is clear to the reader that although I am testing MFI here I agree that it should not be widely used. Since prior authors have used TRMM era satellite precipitation products to estimate MFI, it seems important to at least mention it.

In my opinion, a mixture of satellite products with measured GloREDa would be an ideal and operational solution. That is why the EU Soil Observatory launched also a data collection campaign to get more measured stations data for GloREDa. More info about this call for data in the European Soil Data Centre 2.0 newsletter (February 2023).

I think this is an excellent overall point. I address it more fully above, but I should stress that a combined model is likely to be the best performing output. In this case, the intent of the study is to discuss the value and limitations of a solely IMERG-derived model, but a more general assessment would include diverse input datasets.

Lines 30-35: the cost of soil erosion at global scale, at least for agricultural productivity losses has been estimated to about 8 billion dollars per year. You can find more information in Sartori et al. (2019).

Thank you – an excellent point – I will add this in revision.

In Europe, "Ballabio et al (2017) – Mapping Monthly erosivity in Europe" have developed monthly erosivity maps and datasets based on REDES. Similar has done also for GloREDa at global scale and an article is under preparation to present the monthly erosivity maps. Therefore, measured R-factor data on GloREDa can derive monthly erosivity maps at global scale.

Again – I am very much looking forward to reading the upcoming study and comparing results.

References:

Bezak, N., Mikoš, M., Borrelli, P., Liakos, L. and Panagos, P., 2021. An in-depth statistical analysis of the rainstorms erosivity in Europe. *Catena*, *206*, p.105577.

Matthews, F., Panagos, P. and Verstraeten, G., 2022. Simulating event-scale rainfall erosivity across European climatic regions. *Catena*, *213*, p.106157.

Chen, W., Huang, Y.C., Lebar, K. and Bezak, N., 2023. A systematic review of the incorrect use of an empirical equation for the estimation of the rainfall erosivity around the globe. *Earth-Science Reviews*, p.104339.

---

## Author Response (AR2)

Authors Response: Minor Revision, Emberson 2023 Dynamic Rainfall Erosivity Estimates Derived from GPM IMERG data

Author response in red.

Dear Dr Emberson
The referees have returned positive comments and I believe that the paper is almost ready for publication. Please consider and respond to the minor comments of Referee 2 and submit a final version.
Regards
Graham Jewitt

Dear Dr Jewitt,

Thank you for coordinating a follow-up round of reviews. I have responded to all of the comments from reviewer #2, making all of the suggested changes.

Best,
Robert Emberson

Comments from reviewer #2:

This is my second round of reviewing this manuscript. The author addressed most of the comments. I recommend it for acceptance after a minor revision.

Specific comments:

1.I suggest dividing Figure 5 into two subgraphs, with subgraph A being IMERG derived estimates and subgraph B being REDES, for easy reading and comparison.

Made the suggested change.

2.Lines 258-261: the significant underestimation of IMERG-Final for intense rainfall is the major reason for underestimating rainfall erosivity. The results of Chen et al. (2023) provided evidence: satellite precipitation can accurately capture intense rainfall due to having large probability of detection values (> 0.9), but significantly underestimates rainfall volumes of those captured high-intensity rainfall events. This also indicates that the intense rainfall missed by satellite rainfall products is not the major reason.

Chen, H., Wen, D., Du, Y., Xiong, L., and Wang, L., 2023. Errors of five satellite precipitation products for different rainfall intensities. Atmos. Res. 285, 106622.

Thank you for the reference and comment. I have included this and added the following sentence:
*Recent research has shown that satellite rainfall datasets, including IMERG, may consistently underestimate the total amounts of heavy and storm rainfall [Marc et al. 2022, Chen et al. 2023]*

3.Line 328: please add the specific reference for "Tian and Peters-Lidard, 2010" in the reference section.

Added citation.

4.Line 349: Bezak et al. (2022).

Added parentheses. Thank you for catching that.